# Sequential action of a tRNA base editor in conversion of cytidine to pseudouridine

Satoshi Kimura [1,2,3] ✉, Veerasak Srisuknimit[1,2,3], Kacie L. McCarty[2,4], Peter C. Dedon [5,6], Philip J. Kranzusch [2,4] & Matthew K. Waldor [1,2,3] ✉

Post-transcriptional RNA editing modulates gene expression in a condition-dependent fashion. We recently discovered C-to-Ψ editing in *Vibrio cholerae* tRNA. Here, we characterize the biogenesis, regulation, and functions of this previously undescribed RNA editing process. We show that an enzyme, TrcP, mediates the editing of C-to-U followed by the conversion of U to Ψ, consecutively. AlphaFold-2 predicts that TrcP consists of two globular domains (cytidine deaminase and pseudouridylase) and a long helical domain. The latter domain tethers tRNA substrates during both the C-to-U editing and pseudouridylation, likely enabling a substrate channeling mechanism for efficient catalysis all the way to the terminal product. C-to-Ψ editing both requires and suppresses other modifications, creating an interdependent network of modifications in the tRNA anticodon loop that facilitates coupling of tRNA modification states to iron availability. Our findings provide mechanistic insights into an RNA editing process that likely promotes environmental adaptation.

RNA nucleosides are considerably more diverse than the canonical A, U, G and C nucleosides that correspond to their cognate DNA nucleosides, thereby expanding genetic information and modulating gene expression[1-4]. Some modifications of RNA nucleosides result from processes, such as methylation and acetylation, that add compounds to the base or ribose component of the nucleoside. Additional modifications stem from processes, such as deamination or isomerization that alter the structure of the base. RNA editing, the post-transcriptional conversion of one base to another, accounts for some of the later set of modifications. To date, RNA editing of adenosine into inosine (A-to-I editing) and cytidine into uridine (C-to-U editing) has been described[5,6]. These base conversions are mediated by adenosine deaminases and cytidine deaminases, respectively. In contrast to genomic mutations, RNA editing is regulated, facilitating tissue- and condition-specific control of gene expression[6]. Base conversion diversifies the transcriptome by modulating pre-mRNA splicing patterns as well as the translatome because edited bases can code for

different amino acids[6]. Furthermore, A-to-I editing alters mRNA secondary structure, preventing innate immune responses triggered by mRNAs' rigid double-stranded structures[7]. RNA editing is essential for viability and development in many organisms[8-10].

RNA editing occurs not only in mRNAs but also in non-coding RNAs, such as tRNAs. In bacteria and eukaryotes, A-to-I editing occurs at the wobble position in a subset of tRNAs, thereby expanding their decoding capacity[11-13]. In kinetoplastids and marsupials, RNA editing in the anticodon sequence generates multiple species of tRNAs from a single tRNA gene, enabling subcellular compartment specific translation[14,15]. In addition, tRNA editing is not restricted to the anticodon loop in diverse organisms[16,17] and editing of other regions in tRNA structure is likely associated with structural stability[17]. However, the functions and biosynthesis pathways of tRNA editing have not been well characterized outside of a few model organisms.

A-to-I and C-to-U RNA editing are mediated by zinc-dependent deaminases, the ADAR (adenosine deaminase acting on RNA)[18], ADAT

[1]Division of Infectious Diseases, Brigham and Women's Hospital, Boston, MA, USA. [2]Department of Microbiology, Harvard Medical School, Boston, MA, USA. [3]Howard Hughes Medical Institute, Boston, MA, USA. [4]Department of Cancer Immunology and Virology, Dana-Farber Cancer Institute, Boston, MA, USA. [5]Department of Biological Engineering, Massachusetts Institution of Technology, Cambridge, MA, USA. [6]Singapore-MIT Alliance for Research and Technology Antimicrobial Resistance Interdisciplinary Research Group, Singapore, Singapore. ✉e-mail: s.kimura.res@gmail.com; mwaldor@research.bwh.harvard.edu

(adenosine deaminase acting on tRNAs)[17,19–21], AID/APOBEC[22], and CDA (Cytidine deaminase)[23] families of proteins. These families are distinguishable by their characteristic deaminase domains; the ADAR protein family contains the Adenosine deaminase/Editase domain (A_deamin), whereas the other families possess the Cytidine and deoxycytidylate deaminase domain (CMP_dCMP_dom)[24]. These domains have a consensus sequence that includes zinc-coordinating histidine and/or cysteine residues and a glutamate residue that functions as a proton donor for facilitating the deamination reaction

(Supplementary Fig. 1). All known RNA editing enzymes contain either the A_deamin or the CMP_dCMP_dom domains.

Recently, in studies characterizing the tRNA modification landscape in the cholera pathogen, *Vibrio cholerae*, we discovered C-to-Ψ editing, a RNA editing process in which cytidine is converted into pseudouridine (Ψ)[25]. C-to-Ψ editing was observed at position 32 in the anticodon loop of *V. cholerae* tRNA-Tyr (Fig. 1a). A *V. cholerae* deletion mutant lacking *vca0104* (dubbed *trcP*), which encodes a predicted pseudouridine synthase domain that likely mediates isomerization of

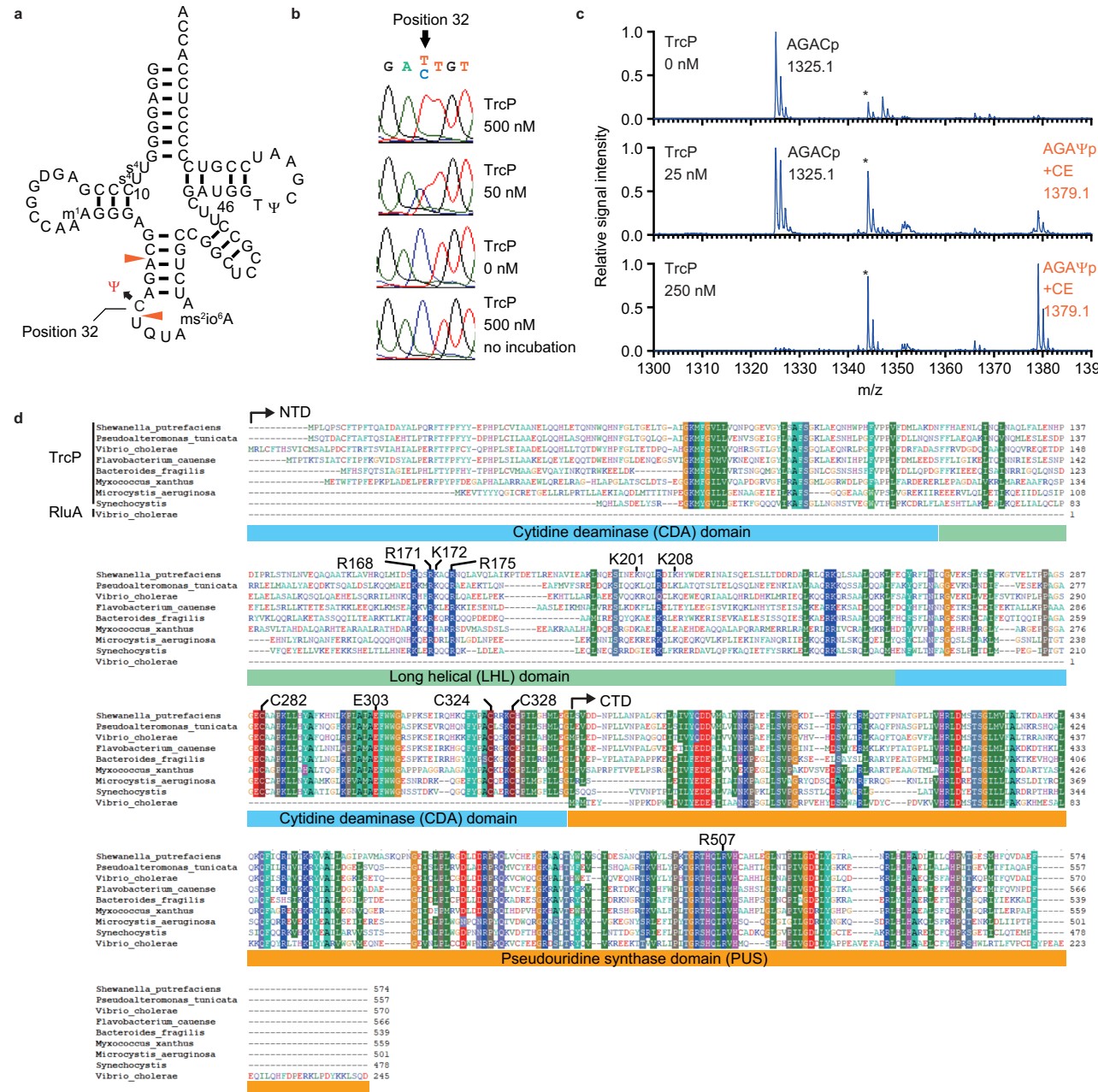

**Fig. 1 | TrcP is sufficient to catalyze C-to-Ψ editing reaction. a** Secondary structure of *V. cholerae* tRNA-Tyr. The C at position 32 is edited to Ψ. **b** Sanger sequencing of amplified cDNA derived from 250 nM tRNAs incubated with recombinant TrcP protein. TrcP concentrations are shown on the right. In the traces, red corresponds to T, and blue corresponds to C. **c** MALDI-TOF analysis of an oligo-protected portion (positions 10–46) of tRNA-Tyr after incubation with recombinant TrcP protein. The tRNA fragment was treated with acrylonitrile to introduce a cyanoethyl group (53 Da) onto Ψ. RNase A digests of a cyanoethylated RNA fragment were subjected to MS. The m/z values and assigned fragments are

shown. Asterisks indicate the fragment corresponding to the tetramer fragment, GAGDp (m/z 1344) derived from positions 13–16. **d** Multiple sequence alignment of TrcP homologs and *V. cholerae* RluA, a paralog of the pseudouridine synthetase domain of TrcP. Similar amino acids that are observed in more than 80% of sequences are shaded with colors. Amino acid residues that are mutated in this study are indicated. TrcP NTD and CTD, and domains assigned by structural modeling, i.e., the cytidine deaminase (CDA) domain (1–114 aa and 250–339 aa), the long helical (LHL) domain (115–249 aa), and the pseudouridine synthase (PUS) domain (351–570 aa) are shown.

U-to-Ψ, was found to be deficient for the entire editing process. However, TrcP lacks the consensus motif shared in deaminases and the mechanisms underlying C-to-Ψ editing and its functional consequences were unclear.

Here we characterize the mechanisms of C-to-Ψ editing. We show that TrcP mediates the stepwise editing of C-to-U followed by the conversion of U to Ψ. Genetic and biochemical studies revealed that TrcP relies on its two catalytic domains, including a cytidine deaminase domain and a pseudouridine synthase domain, for C-to-Ψ editing. AlphaFold 2-based structural modeling uncovered a third unique long helical domain within TrcP that likely binds and orients the substrate tRNA during both reactions, suggesting TrcP mediated C-to-Ψ editing relies on a substrate channeling mechanism. Additional studies of the control of C-to-Ψ editing uncovered a modification circuit in the tRNA anticodon loop whereby C-to-Ψ editing at position 32 requires methylthiolation of A at position 37 as $ms^2io^6A$ and suppresses the conversion of G to Q at wobble position 34. Functionally, this modification circuit facilitates coupling of tRNA modification states to environmental iron availability and impacts decoding, likely for promoting environmental adaptation.

## Results

### TrcP mediates C-to-Ψ conversion

Genetic analyses established that *vca0104* (*trcP*), is required for the conversion of C-to-Ψ in the *V. cholerae* tRNA-Tyr[25]. To address whether TrcP is sufficient to catalyze C-to-Ψ editing, an in vitro editing assay was carried out using purified recombinant TrcP protein and an unedited tRNA-Tyr isolated from a *ΔtrcP* strain (Supplementary Fig. 2). In this assay, editing frequency was measured by Sanger sequencing of cDNA derived from the tRNA-products of the reaction. Note that in Sanger sequencing, both U and Ψ yield a T. Without addition of TrcP, the unedited tRNA-Tyr was read as a C, whereas addition of increasing amounts of TrcP to the reaction increased the frequency of T and decreased the frequency of C at position 32 of tRNA-Tyr (Fig. 1b). These data indicate that TrcP catalyzes C-to-U or C-to-Ψ conversion.

Cyanoethyl labeling coupled with MALDI-TOF mass spectrometry (MALDI-TOF-MS) was used to distinguish between U and Ψ[25,26]. After incubation with recombinant TrcP a fragment of tRNA-Tyr (position 10–46) was purified and treated with acrylonitrile, which specifically attaches a cyanoethyl group on Ψ. MALDI-TOF-MS analyses of RNA fragments generated by RNase A digestion revealed that the unmodified fragment (AGACp; m/z 1325.1) was increasingly converted to the cyanoethylated fragment (AGAΨp + CE; m/z 1379.1) as the TrcP concentration increased (Fig. 1c). Together these observations establish that TrcP catalyzes C-to-Ψ base editing.

### Phylogenetic distribution of TrcP

Analysis of the TrcP primary amino acid sequence with the NCBI Conserved Domain Search tool revealed it contains two recognizable domains. Its C-terminus (CTD, amino acids 351–570) encodes a RluA family pseudouridine synthase (PUS) domain (E-value 5.43 e-74), which includes several highly conserved residues implicated in catalysis (such as R507), that mediates U to Ψ conversion[27] (Fig. 1d). Its N-terminus (NTD, amino acids 1–350) includes a region that has modest similarity to the Smc super family (amino acids 132–258; E-value 7.32 e-6). BLAST searches were carried out to probe the phylogenic distribution of TrcP. When the complete amino acid of TrcP was used as the query, the search only yielded proteins that contain a PUS domain without sequences corresponding to TrcP's NTD. However, when TrcP's NTD was the search query, TrcP homologs in other Vibrionaceae, and genera related to vibrios, including *Shewanella* and *Pseudoalteromonas*, became apparent; in addition, certain species in phyla that are phylogenetically distant from *V. cholerae*, such as *Bacteroides*, *Nostoc*, *Synechococcus*, and *Myxococcus* also encode TrcP homologs (Fig. 1d and Supplementary Fig. 3), suggesting that though

relatively uncommon, potential TrcP-like C-to-Ψ base editors are broadly distributed among bacteria. Notably, a multiple sequence alignment of TrcP homologs showed that they all contain sequences similar to TrcP's NTD and CTD (Fig. 1d). The co-occurrence of both the NTD and PUS domains in TrcP homologs suggests that these two domains function together to mediate C-to-Ψ conversion.

### TrcP's two domains mediate C-to-U and U-to-Ψ conversion

Since TrcP's C-terminal PUS domain likely converts U to Ψ, we hypothesized that C-to-Ψ conversion by TrcP is accomplished by sequential C-to-U and U-to-Ψ conversion mediated by its NTD and CTD respectively (Fig. 2a). To test this idea, we purified recombinant TrcP-NTD, TrcP-CTD, and a TrcP mutant in which a conserved catalytic arginine residue in the PUS domain was mutated to alanine (R507A)(Fig. 2b and Supplementary Fig. 2). The catalytic activity of these proteins was evaluated by first incubating them in vitro with the unedited substrate tRNA-Tyr isolated from a *ΔtrcP* strain, and then analyzing the products by MALDI-TOF MS coupled with cyanoethylation as described above. The TrcP-NTD fully converted the cytidine at position 32 into uridine but not to pseudouridine (Fig. 2c), indicating that the TrcP-NTD is a cytidine deaminase sufficient for C-to-U base editing but not for U-to-Ψ conversion. The TrcP-CTD did not catalyze C-to-U conversion, supporting the idea that the NTD is necessary for C-to-U conversion (Fig. 2c). Like the TrcP-NTD, the R507A PUS domain mutant, catalyzed C-to-U editing but failed to convert uridine to pseudouridine (Fig. 2c), confirming that the intact NTD is sufficient for base editing and suggesting that the predicted catalytic site in the TrcP-CTD is necessary for pseudouridylation. Together, these results support the idea that TrcP catalyzes C-to-Ψ conversion in a stepwise reaction mechanism, where C is first converted to U, and then U is converted to Ψ.

The capacity of the TrcP-CTD to catalyze U-to-Ψ conversion on tRNA-Tyr U32 appears to depend on its linkage to TrcP's NTD. WT TrcP catalyzed pseudouridylation of tRNA-Tyr U32 generated by TrcP-NTD (Fig. 2c, NTD → TrcP), but TrcP-CTD did not (Fig. 2c, NTD → CTD). Furthermore, U-to-Ψ conversion was not observed when the TrcP-CTD and the TrcP-NTD were added to a reaction mixture as separate proteins (Fig. 2c, NTD + CTD). We reasoned that TrcP's CTD must be coupled to its NTD for pseudouridylation because the NTD facilitates substrate tRNA binding and/or stabilizes the proper folding of the CTD.

### TrcP contains two globular domains and a long helical domain

ColabFold[28], a version of the structure prediction algorithm AlphaFold-2[29] was used to gain further insights into the mechanism of C-to-Ψ editing by TrcP. The program yielded five nearly identical high confidence structural models (LDDT generally >80) (Supplementary Fig. 4). These models all suggest that TrcP is composed of two globular domains along with a long helical (LHL) domain consisting of three long helices (Fig. 3a). The TrcP-CTD forms a globular domain, strongly suggesting that this domain corresponds to the PUS domain. The TrcP-NTD includes the other globular domain with the insertion of the long helical domain.

Additional structural homology searches using Dali[30] strongly suggest that the globular domain in the TrcP-NTD corresponds to a cytidine deaminase (CDA) domain. When the full-length TrcP-NTD, containing the globular domain and the LHL domain, was used as a query to identify structurally similar proteins, most of the hits were proteins that contain long helices, such as cohesin and membrane proteins (Supplementary Data 1). However, when the query was the sequence corresponding to the TrcP-NTD globular domain without the LHL domain (Fig. 3a and Fig. 1d), the search identified several proteins annotated as deaminases (Supplementary Data 2). These hits included deaminases targeting cytidine such as blasticidin-S deaminase (PDB 3oj6) and cytidine deaminase (PDB 4eg2). The structure of Blasticidine-S deaminase (BSD) is highly similar to the predicted structure of TrcP-NTD globular domain (Fig. 1d, Fig. 3b and

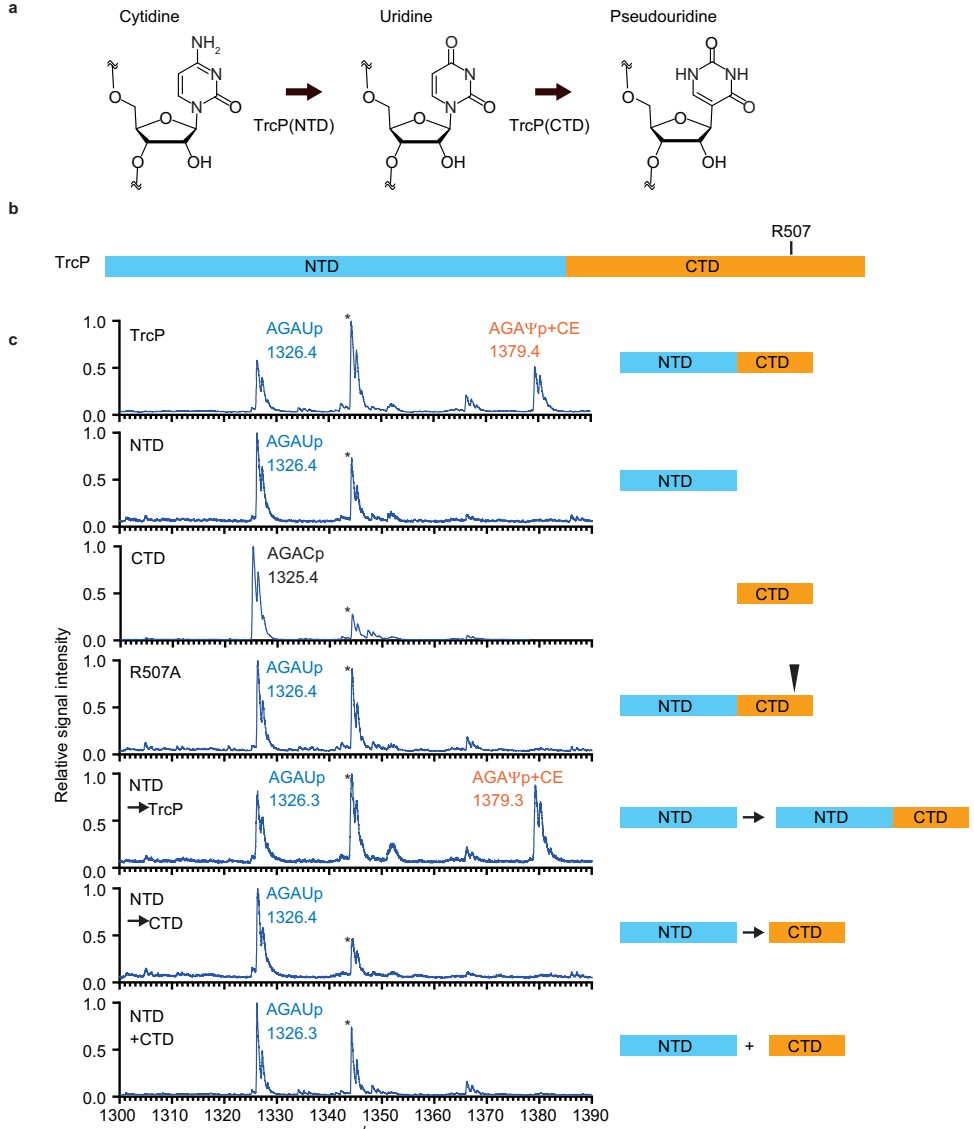

**Fig. 2 | TrcP includes a C-to-U RNA editor and a pseudouridylase. a** Schematic of the C-to-Ψ editing reaction by TrcP. **b** TrcP domain structure with the position of R507. **c** MALDI-TOF analysis of an oligo-protected portion (positions 10–46) of tRNA-Tyr after incubation with recombinant TrcP and mutant derivatives. The tRNA fragment was treated with acrylonitrile to introduce a cyanoethyl group (53 Da) to Ψ. RNase A digests of a cyanoethylated RNA fragment were subjected to MS. The m/z values and assigned fragments are shown. Asterisks indicate the fragment corresponding to the tetramer fragment, GAGDp (m/z 1344) derived from positions 13–16.

Supplementary Fig. 5). Together these finding are consistent with the idea that the TrcP-NTD globular domain encodes a cytidine deaminase. Notably, the deaminase proteins identified in the structural search were not identified in the BLAST search with the TrcP-NTD sequence, raising the possibility that the TrcP-CDA domain evolved independently of other deaminase domains, including known C-to-U RNA editors such as CDAT8 in archaea[16] and ADAT2/ADAT3 in Trypanosoma[31].

**Reaction mechanism of C-to-U editing by the TrcP CDA domain**

Deaminases that target adenine and cytosine rely on a zinc ion as a co-factor to activate a substrate water molecule[32]. In these enzymes, the zinc ion is usually coordinated by three cysteine residues or one histidine and two cysteine residues[33] (Supplementary Fig. 1). In the structural model of the TrcP CDA domain there are three conserved cysteine residues, C282, C324, and C328 (Fig. 1d) that form a cluster with spatial coordinates (Fig. 3b) very similar to those of the Zn-coordinating cysteines in blasticidin-S deaminase (Fig. 3b and

Supplementary Fig. 5), suggesting that the TrcP-CDA also utilizes zinc ion as a co-factor. To assess this possibility, we measured the zinc concentration in TrcP. Without denaturation, only trace amount of free zinc was detected in TrcP solution (0.91 μM in 49 μM TrcP), whereas with acid denaturing, free zinc ion was detected in TrcP solution (27.9 μM in 49 μM TrcP) (Fig. 3c). An in vivo complementation assay was carried out to assess the requirement for these conserved cysteines for C-to-U editing. A Δ*trcP* mutant strain was transformed with vectors encoding cysteine mutants and the editing frequency was measured by Sanger sequencing. No editing activity was observed for any of the three cysteine substitution mutants even though all three mutant proteins were expressed at levels similar to the WT protein (Fig. 3d and Supplementary Fig. 6). Together these observations suggest that zinc is coordinated by a cysteine cluster in the TrcP CDA catalytic site and the absence of any of these residues impairs TrcP's capacity for C-to-U editing.

Known nucleoside base deaminases rely on a glutamate in the catalytic pocket to facilitate the deamination reaction[34]. Like BSD, the

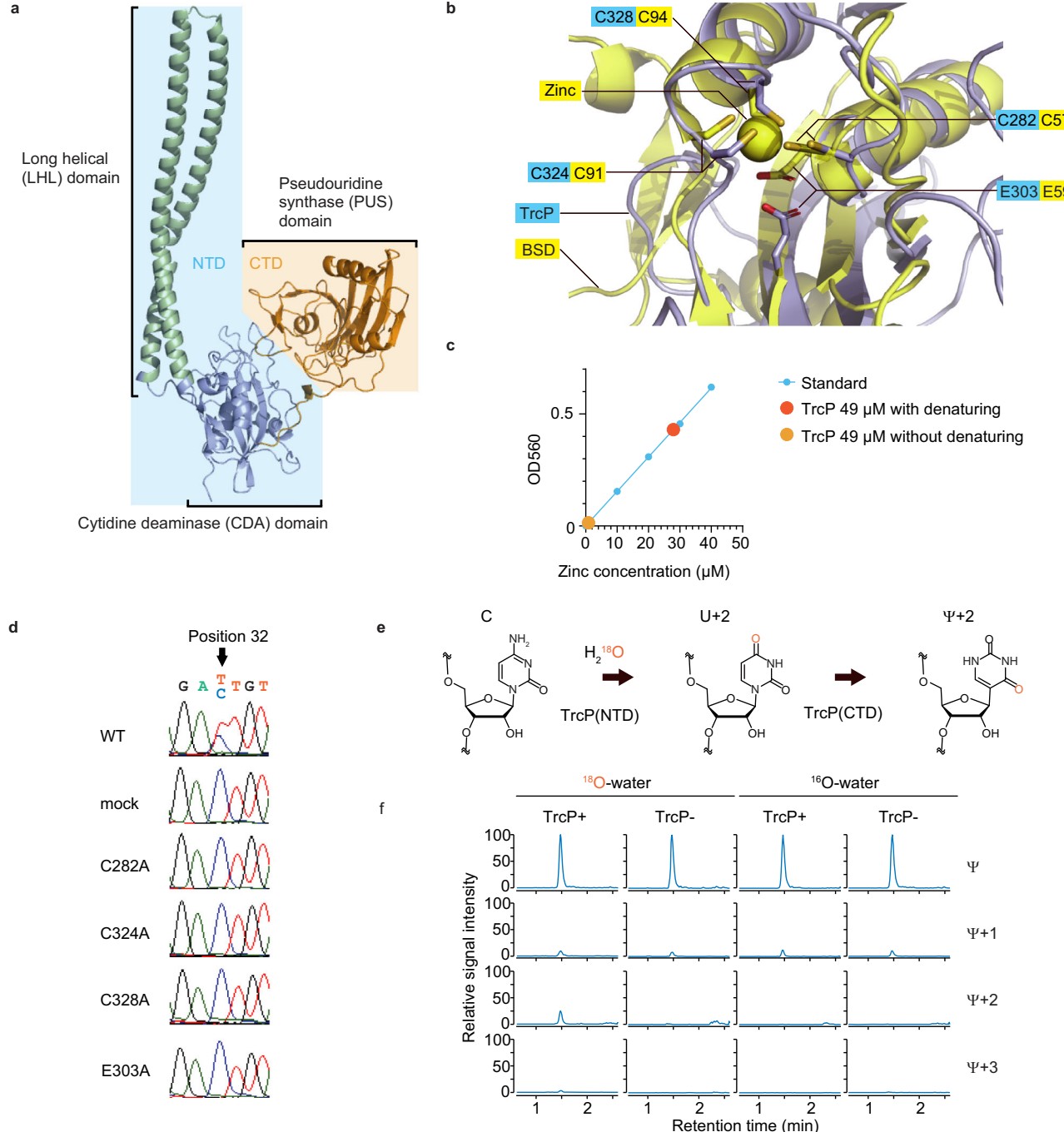

**Fig. 3 | Reaction mechanisms of TrcP catalyzing C-to-U editing. a** A structural model of TrcP protein generate by ColabFold[28]. TrcP is composed of two globular domains, the cytidine deaminase (CDA) domain (light blue) and pseudouridine synthetase (PUS) domain (orange), and a long helical (LHL) domain (green). NTD and CTD are shaded with light blue and light yellow, respectively. **b** Alignment of the predicted catalytic site in the TrcP CDA domain (light blue) with the Blasticidin-S deaminase (BSD) (yellow). The residues of BSD and its coordinating zinc ion are shown in the yellow background, whereas the residues of TrcP are shown in the light blue background. Oxygen and sulfur residues are also colored red and gold, respectively. **c** Colorimetric measurement of zinc concentration of TrcP solution.

Source data are provided as a Source Data File. **d** In vivo complementation analysis of TrcP mutants. Sanger sequencing results of tRNA-Tyr cDNA derived from RNA isolated from *trcP* knockout strains expressing TrcP (WT) or the indicated mutant derivatives. **e** A schematic of the C-to-Ψ editing reaction by TrcP. TrcP's NTD uses water as a source of oxygen for C-to-U editing. In U and Ψ, the expected positions of [18]O-labeled oxygen from water are shown in red. **f** Nucleoside analysis of tRNAs incubated with or without TrcP and stable isotope-labeled water ([18]O-water). The reaction conditions and the detected nucleosides are shown above and right, respectively.

TrcP CDA contains a conserved glutamate within its catalytic pocket (Figs. 1d, 3b and Supplementary Fig. 5). This E303 residue was found to be critical for C-to-U editing in the in vivo complementation assay (Fig. 3d and Supplementary Fig. 6). Catalytic carboxyl groups of E59 and E303 are located within the active sites of BSD and TrcP respectively (Fig. 3b), suggesting that the TrcP-CDA also relies on glutamate

to facilitate the cytidine deamination reaction. Collectively, our findings suggest that even though the TrcP CDA domain and BSD lack primary amino acid sequence similarity, their respective catalytic sites both include a zinc ion coordinated by a cluster of three cysteine residues and a glutamate residue. All four of these residues are also present in the homologs identified in the TrcP-NTD BLAST search

(Fig. 1d and Supplementary Fig. 1), suggesting that these proteins also correspond to unrecognized cytidine deaminases.

During C-to-U editing, an enamine group in cytosine is converted to a carbonyl group (Fig. 3e). This process requires an external source of oxygen and characterized nucleoside base deaminases use water as the oxygen source[32,34]. An in vitro editing assay with water containing [18]O, which is 2 Da heavier than naturally abundant [16]O, was used to test whether TrcP also uses water as an oxygen source (Fig. 3e). When the reaction mixture included 50% [18]O-labeled water, we observed the heavier Ψ (Ψ+2) (Fig. 3f), indicating that labeled oxygen was incorporated into Ψ. Detection of this labeled product required the addition of TrcP protein and [18]O to the reaction (Fig. 3f). Thus, TrcP uses water as an oxygen source in the deamination process in C-to-Ψ editing.

## The TrcP long helical domain facilitates tRNA-Tyr binding

TrcP's two globular domains appear to catalyze C-to-Ψ editing. We speculated that the enzyme's long helical (LHL) domain facilitates substrate tRNA binding. To predict potential RNA binding regions in TrcP, we calculated the surface potential in the model structures, since positively charged regions can bind to the negatively charged RNA mainchain. Three positively charged TrcP regions were identified: two regions in the pockets of the globular domains and one in a patch near the tip of the long helical domain (Fig. 4a). To assess the possibility that the positively charged patch in the LHL domain is involved in substrate tRNA binding, a mutant TrcP (KR mutant) in which six arginine and lysine residues within the positively charged patch (R168, R171, K172, R175, K201, and K208) were substituted for alanine was created. The KR mutant did not have C-to-U editing activity (Fig. 4b and Supplementary Fig. 6). The KR mutant had the same melting temperature as WT TrcP protein, suggesting that the six substitutions did not affect overall protein folding (Supplementary Fig. 8). Furthermore, single substitution mutants (R168A, R175A, K201A, K208A) were also generated and the mutant proteins only had marginal C-to-U editing activities in the in vivo complementation assay (Supplementary Figs. 6 and 7). Together, these findings strongly support the idea that the positively charged patch in the LHL domain is required for TrcP editing. The binding capacity of TrcP to substrate tRNAs isolated from the Δ*trcP* strain was assessed with a gel mobility shift assay. As the concentration of WT TrcP protein was increased, a shifted band that presumably represents a complex of TrcP protein and tRNA-Tyr was observed; in contrast, a shifted fragment was not detected when the same concentration of the TrcP-KRmutant was used (Fig. 4c). Furthermore, TrcP appears to specifically bind tRNA-Tyr, since a shifted band was not observed when tRNA-Asp was incubated with the WT TrcP protein (Fig. 4c).

The TrcP positively charged patch in its LHL domain is predicted to be relatively distant from the catalytic centers of both the CDA and PUS domains, suggesting that this patch binds to the upper body of the tRNA-Tyr molecule. To assess which parts of the tRNA-Tyr structure are required for TrcP editing activity, we conducted in vitro editing assays with mutant tRNAs that lacked the CCA end (ΔCCA), acceptor arm (ΔACarm), T-arm (ΔTarm), D-arm (ΔDarm), or variable loop (ΔVloop) (Supplementary Fig. 9). The tRNA substrates used in these reactions were generated by in vitro transcription and therefore lacked modifications. Partial C-to-U editing was observed with full-length tRNA-Tyr and the mutant lacking the CCA end, whereas the other four mutant tRNA-Tyr substrates were not edited (Fig. 4d). These observations suggest that the properly folded structure of tRNA-Tyr is necessary for editing by TrcP.

Next, we asked whether the positively charged patch in the LHL domain facilitates TrcP's pseudouridylation activity. For these experiments, substrate tRNA-Tyr isolated from the Δ*trcP* strain was initially incubated with TrcP-NTD and [18]O-labeled water, yielding tRNA-Tyr with [18]O-labeled uridine at position 32. The [18]O-labeled tRNAs were then reacted with WT or KRmutant TrcP and the amount

of labeled uridine was tracked (Fig. 4e). WT TrcP converted a larger fraction of the labeled uridine to pseudouridine than the KRmutant (Fig. 4f), indicating that KRmutant has less pseudouridylation activity than WT TrcP. This observation strongly suggests that the positively charged patch in the LHL domain promotes TrcP pseudouridylation activity and at least partly accounts for the lower activity of the CTD vs full-length TrcP in isomerization of U to Ψ (Figs. 2c and 4f).

To corroborate the hypothesis that the TrcP-NTD facilitates the pseudouridylation activity of the CTD by promoting substrate binding, we created a chimeric enzyme. The TrcP-NTD was fused to RluA, a *V. cholerae* pseudouridylase that does not ordinarily target tRNA-Tyr for pseudouridylation (Fig. 4f). However, the chimeric TrcP-NTD-RluA protein had pseudouridylation activity on tRNA-Tyr at position 32 (Fig. 4f). We speculate that tethering of the tRNA-Tyr substrate by the LHL domain facilitates RluA's pseudouridylation activity.

## An iron-responsive modification network in tRNA-Tyr

The synthesis of certain tRNA modifications depends on the presence or absence of other modifications, likely due to substrate recognition by modification enzymes[35,36]. In vitro transcribed (unmodified) tRNA-Tyr was used to test whether modifications in tRNA-Tyr influence the efficiency of C-to-Ψ editing. TrcP C-to-Ψ editing of an unmodified in vitro transcribed substrate was less efficient than on partially modified tRNAs extracted from the Δ*trcP* strain; even with 250 nM TrcP and 250 nM of unmodified tRNA-Tyr, less than half of position 32 cytidine was converted to pseudouridine, whereas all of the cytidine 32 was converted to pseudouridine when the substrate tRNA-Tyr was derived from Δ*trcP V. cholerae* (compare Fig. 5a and Fig. 1c). This observation suggests that other modifications within tRNA-Tyr facilitate TrcP C-to-Ψ editing.

We suspected that modifications in the anticodon could modulate the efficiency of C-to-Ψ editing because in eukaryotes modifications at position 32 often depend on other modifications in the anticodon loop[35]. Besides the Ψ at position 32, there are two additional modifications, at positions 34 (queuosine, Q) and 37 (ms²io⁶A), in the anticodon loop of tRNA-Tyr (Supplementary Fig. 10). To test whether these modifications influence C-to-Ψ editing, editing in total tRNAs extracted from mutant strains lacking the enzymes that create these modifications was assessed. At position 37, ms²io⁶A is synthesized in three steps: (1) addition of an isopentenyl group to adenine by MiaA, (2) methyl-thiolation of i⁶A by MiaB, and (3) hydroxylation of ms²i⁶A by MiaE (Supplementary Fig. 10a). No editing of tRNA-Tyr was observed in tRNA prepared from either Δ*miaA* or Δ*miaB* mutant strains but editing of RNA derived from the Δ*miaE* strain was detected (Fig. 5b), suggesting modification of adenine at position 37 by both an isopentenyl and a methyl-thio group are required for TrcP C-to-Ψ editing at position 32 of tRNA-Tyr. Since the isopentenyl group is required for methyl-thiolation by MiaB in *E. coli*[37], our findings strongly suggest that the methyl-thio group in ms²i⁶A is critical for TrcP C-to-Ψ editing. An in vitro editing assay showed that tRNA-Tyr with ms²i⁶A37 (Δ*trcP*) is a better substrate than tRNA-Tyr containing io⁶A37 (Δ*miaB*), consistent with the idea that the methyl-thio group facilitates the TrcP editing reaction (Fig. 5c). Note that in the presence of an excess amount of TrcP protein, tRNA-Tyr from the Δ*miaB* strain can be edited, suggesting that methyl-thio group is not essential for editing. Conversely, the frequency of editing was not altered by the absence of queuosine at position 34 when tRNA from the *tgt* mutant strain was tested (Fig. 5b), indicating that the presence of the modified Q34 does not influence TrcP editing.

The observation that TrcP C-to-Ψ editing is dependent on ms²io⁶A introduced the possibility that editing is modulated by environmental conditions, particularly iron availability because generation of ms²io⁶A in *E. coli* is controlled by the availability of this nutrient[38]. To test this hypothesis, we analyzed the editing frequency in RNA derived from

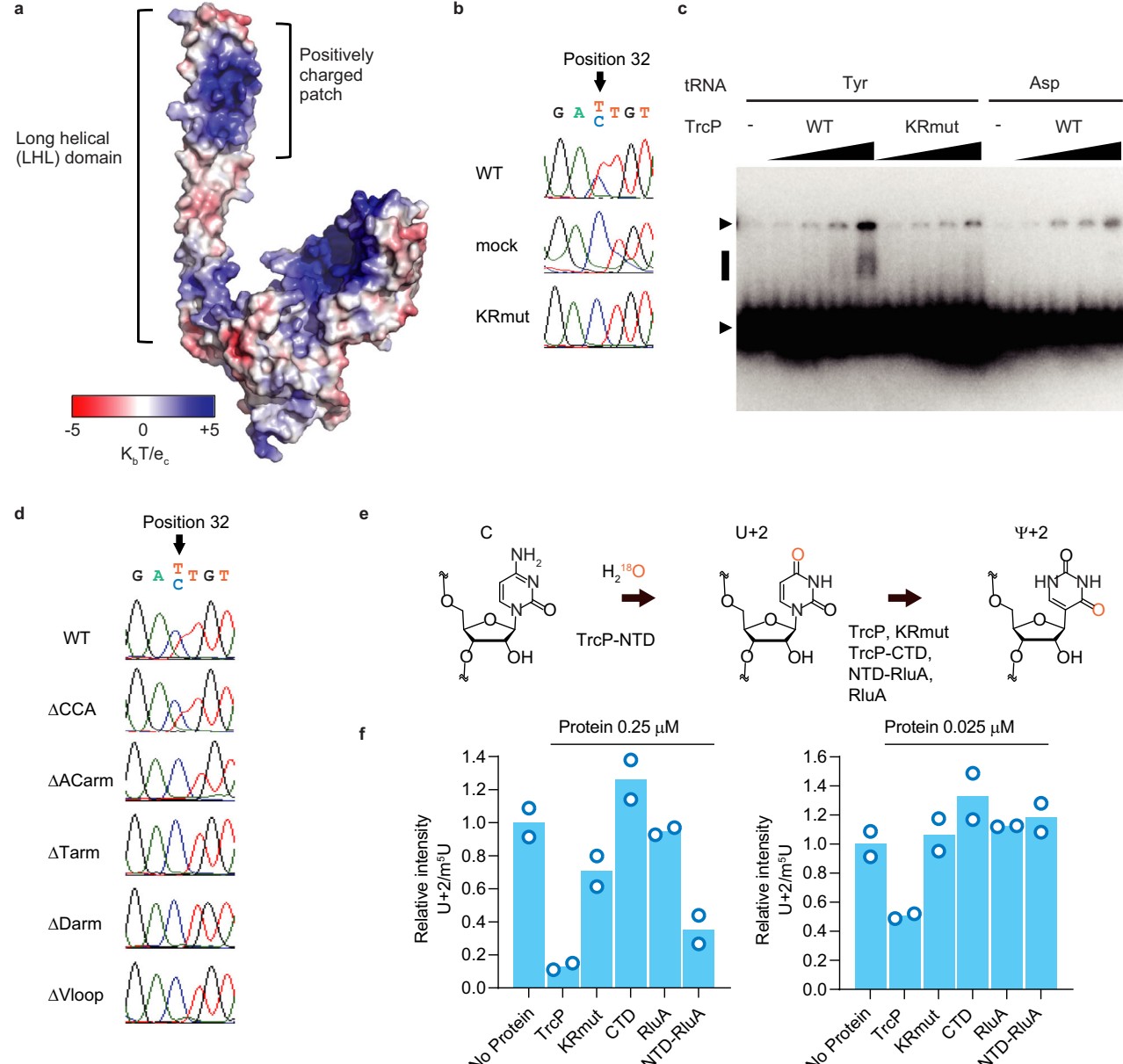

**Fig. 4 | The long helical domain facilitates substrate tRNA binding. a** Predicted surface electrostatic potential of the structural model of TrcP. The color key is shown below (unit $K_bT/e_c$). **b** In vivo complementation analysis of the KR mutant. Sanger sequencing results of tRNA-Tyr cDNA derived from RNA isolated from *trcP* knockout strains expressing TrcP (WT) or mutant derivatives. **c** Gel mobility shift assay with TrcP (WT and KR mutant) and tRNAs (tRNA-Tyr and tRNA-Asp). Upper and lower arrowheads represent wells and free tRNA signals, respectively. The black line indicates the shifted band, presumably representing the soluble tRNA-TrcP complex. Protein concentrations were 0, 0.1, 1, 2, and 5 µM, and the tRNA concentration was 1 µM. An uncropped blot is provided in Source Data. Representative data from two independent experiments with similar results is shown. **d** Sanger sequencing of cDNA derived from 250 nM tRNA-Tyr (WT) and the indicated mutants incubated with 750 nM recombinant TrcP protein. The secondary structures of mutant tRNAs are shown in Supplementary Fig. 9. **e** A schematic of the experiment in f. TrcP-NTD was used for generating $^{18}$O-labeled U32 in tRNA-Tyr. Purified tRNA-Tyr was then incubated with indicated enzymes. **f** Nucleoside analysis of tRNA-Tyr after the reaction with the indicated proteins. Conversion of U into Ψ was assessed by the signals of $^{18}$O-labeled U (U + 2). The signals were normalized by m$^5$U. The bar represents the average values of two independent reaction results represented by the points. This experiment was performed once. Source data are provided as a Source Data File.

cells grown in iron replete or iron depleted conditions, generated with the iron-chelator, 2′, 2′-dipyridyl (dip). As the concentration of dip increased, a decrease in editing frequency was observed (Fig. 5d). In the presence of >175 µM dip, editing was completely eliminated. The A-to-I editing frequency at the wobble position in tRNA-Arg2C did not change with iron depletion (Supplementary Fig. 11), suggesting that iron concentration does not generally modulate editing but specifically controls C-to-Ψ editing. Together these observations reveal that TrcP editing of C-to-Ψ at position 32 is controlled by iron availability

likely because efficient editing in the cell relies on iron-controlled methyl-thio modification at position 37.

Since we observed that modification of position 37 to ms$^2$io$^6$A modulates the frequency of C-to-Ψ editing, we wondered if editing at position 32 can affect the synthesis of other modifications. To investigate this possibility, tRNA-Tyr from WT and Δ*trcP* strains were harvested from log and stationary phase cultures and liquid chromatography mass spectrometry (LC-MS) was used to analyze nucleoside modifications. In log phase samples, there were lower

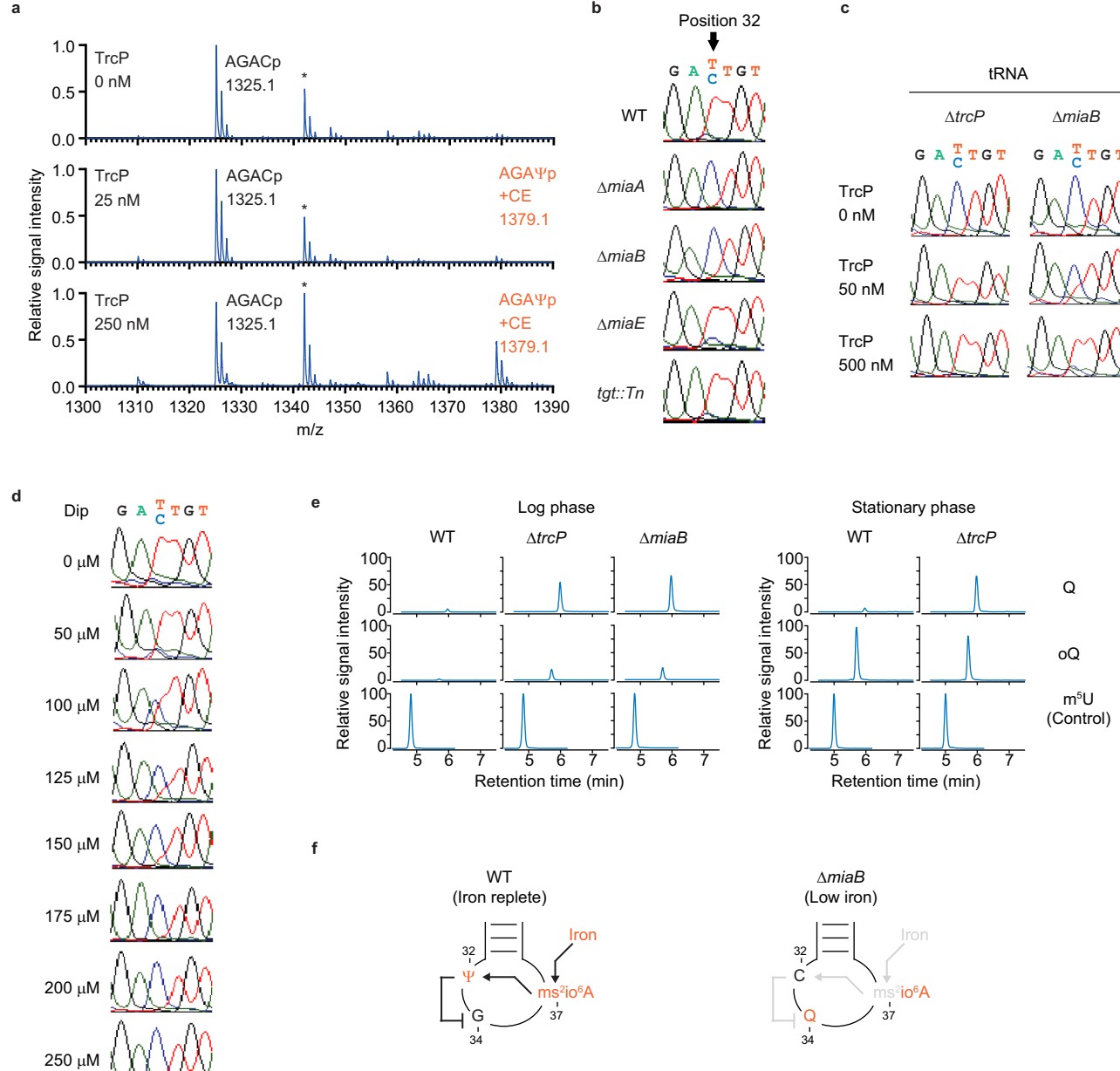

**Fig. 5 | An iron-responsive modification network in *V. cholerae* tRNA-Tyr.**
**a** MALDI-TOF analysis of the oligo-protected portion (positions 10–46) of in vitro transcribed tRNA-Tyr. tRNA (250 nM) was reacted with 0 nM (top), 25 nM (middle), and 250 nM (bottom) recombinant TrcP. Oligo-protected portions were incubated with acrylonitrile, which specifically cyanoethylates (CE) pseudouridine, increasing its mass by 53 Da. m/z values and assigned fragment sequences are shown. Asterisks indicate the fragment corresponding to the tetramer fragment, GAGDp (m/z 1344) derived from positions 13–16. **b** Sanger sequence of tRNA-Tyr cDNA from RNA isolated from WT (Top), Δ*miaA* (second top), Δ*miaB* (middle), Δ*miaE* (second bottom), and *tgt::Tn* (bottom) strains. **c** Sanger sequence of cDNA of tRNA-Tyr isolated from a Δ*trcP* strain (left) or a Δ*miaB* strain (right). tRNAs (250 nM) were incubated with 0 nM, 50 nM, and 500 nM TrcP. **d** Sanger sequence of tRNA-Tyr cDNA from RNA isolated from the WT strain cultured with or without dipyridyl (dip) at the indicated concentrations. **e** Nucleoside analysis of tRNA-Tyr from WT, Δ*trcP* and Δ*miaB* strains. The left panel shows log phase results, and the right panel shows stationary phase results. The detected nucleosides are shown on the right. The signal intensity is normalized with m5U signals in each sample. Representative data from two independent experiments with similar results is shown. **f** Schematic of proposed interdependency network of modifications in the tRNA-Tyr anticodon loop. In WT under iron-replete conditions, ms2io6A37 facilitates C-to-Ψ editing at position 32, which in turn suppresses Q formation at position 34 (Left). In contrast, the absence of *miaB* or low iron conditions eliminates methyl-thio modification at position 37, suppressing C-to-Ψ editing and thereby relieving inhibition of Q34 biogenesis (Right).

signals of Q and its precursor oQ in the WT strain than those in the Δ*trcP* strain (Fig. 5e and Supplementary Fig. 10b). The elevated signals of Q and oQ in the Δ*trcP* strain were also observed in Δ*miaB* strain, where C-to-Ψ editing is also eliminated (Fig. 5e). In stationary phase samples, the oQ signal level was similar in the two strains, whereas the Q signal remained low in the WT (Fig. 5e). These data strongly suggest that the biogenesis of Q at position 34 is impaired by TrcP's C-to-Ψ editing (Fig. 5f).

## C-to-Ψ editing promotes Tyr decoding
Finally, we assessed the effects of TrcP derived C-to-Ψ editing on the decoding capacity of tRNA-Tyr to begin to investigate the function of editing. A reporter system in which the translation efficiency of specific codons is evaluated with the efficiency of frameshifting was engineered[39,40] (see Methods, Fig. 6a). The reporter system was introduced into the WT, Δ*trcP*, Δ*tgt*, and Δ*trcP*/Δ*tgt* strains to assess the decoding efficiency of two tyrosine codons in log and early stationary

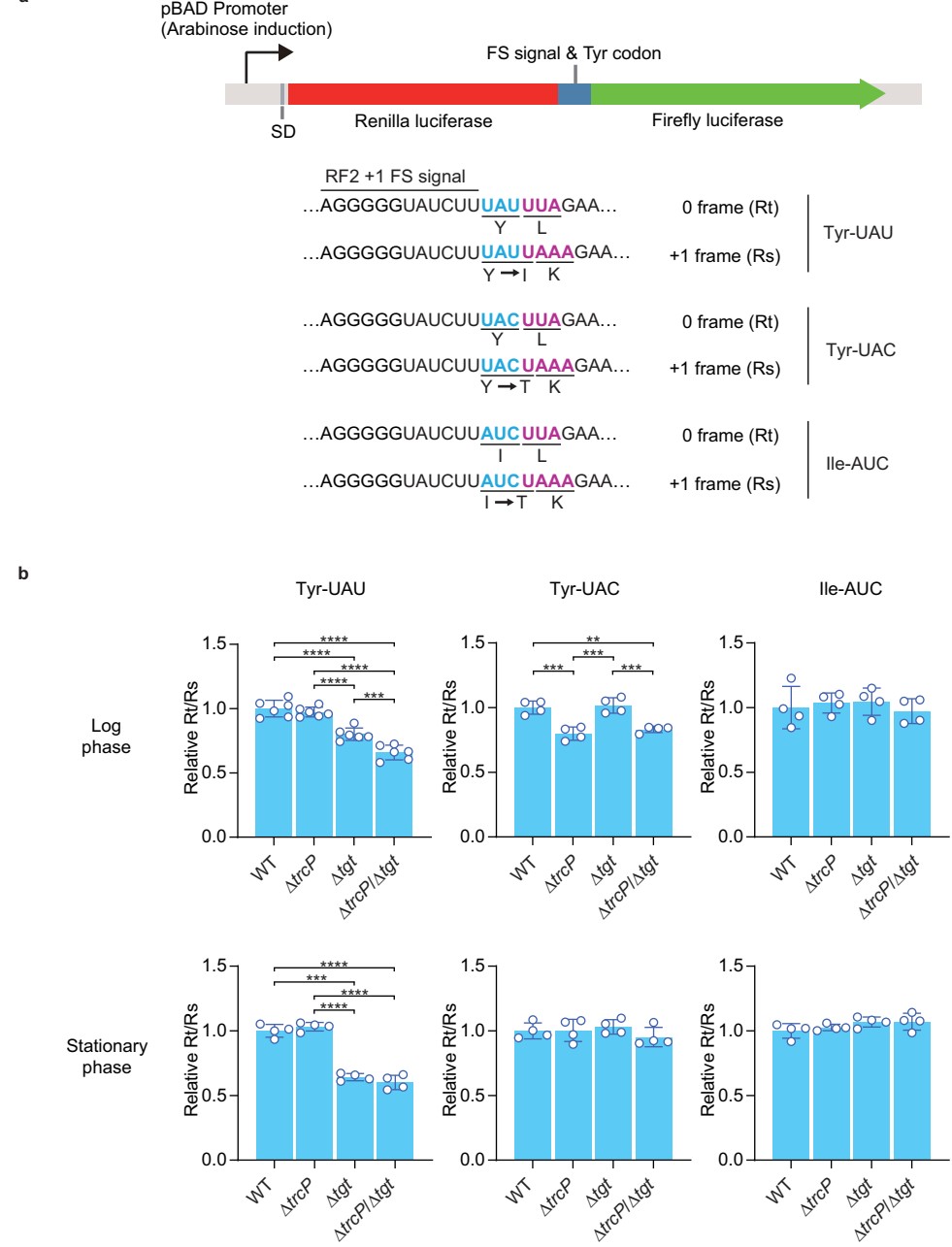

**Fig. 6 | C-to-Ψ editing facilitates Tyr decoding. a** Schematic of reporter construct for measuring decoding ability of tRNA-Tyr and tRNA-Ile. **b** Reporter assays evaluating decoding of Tyr and Ile codons. Relative Rt/Rs value (see methods) is indicated. The top and bottom panels show the results derived from log and stationary phase cultures, respectively. The tested codons are shown above, and the tested strains are labeled below. Ordinary one-way ANOVA was used for a statistical test. Comparisons were made in all combinations among the strains, and Tukey was used for correction for multiple comparisons ($^{**}p < 0.01$; $^{***}p < 0.001$; $^{****}p < 0.0001$). The bar represents the average values of biologically independent culture results represented by the points. $n = 6$ for log phase UAU reporter assay and $n = 4$ for the other conditions. The standard deviations (SD) were shown as error bars. Source data with exact $p$ values are provided as a Source Data File.

phase. We found that editing influenced decoding in both a codon specific and growth phase specific fashion. In log phase, absence of *trcP* suppressed decoding of UAC but not of UAU (p-value 0.0004) (Fig. 6b), whereas absence of *tgt* decreased UAU decoding (p-value < 0.0001) but not UAC decoding as reported in *Salmonella*[39] (Fig. 6b). In the absence of both *tgt* and *trcP*, UAC decoding was similar to that observed in the absence of *trcP* only, but UAU decoding was even more reduced than in the single *tgt* knockout strain (p-value 0.0009) (Fig. 6b), suggesting that C-to-Ψ editing at position 32 facilitates UAU decoding along with Q modification of position 34. In stationary phase, the absence of *trcP* did not affect decoding of either UAU or

UAC. In both growth phases, neither the absence of *trcP* or *tgt* affected AUC decoding, demonstrating the specificity of editing and Q modification in decoding. Thus, several factors including growth conditions determine whether C-to-Ψ editing facilitates Tyr codon decoding.

## Discussion

Here, we characterized the biogenesis of C-to-Ψ RNA editing. A single enzyme, TrcP, was found to be sufficient to catalyze the conversion of C-to-Ψ. Examples of nucleosides undergoing both editing and modification, such as A-to-I and C-to-U editing followed by methylation,

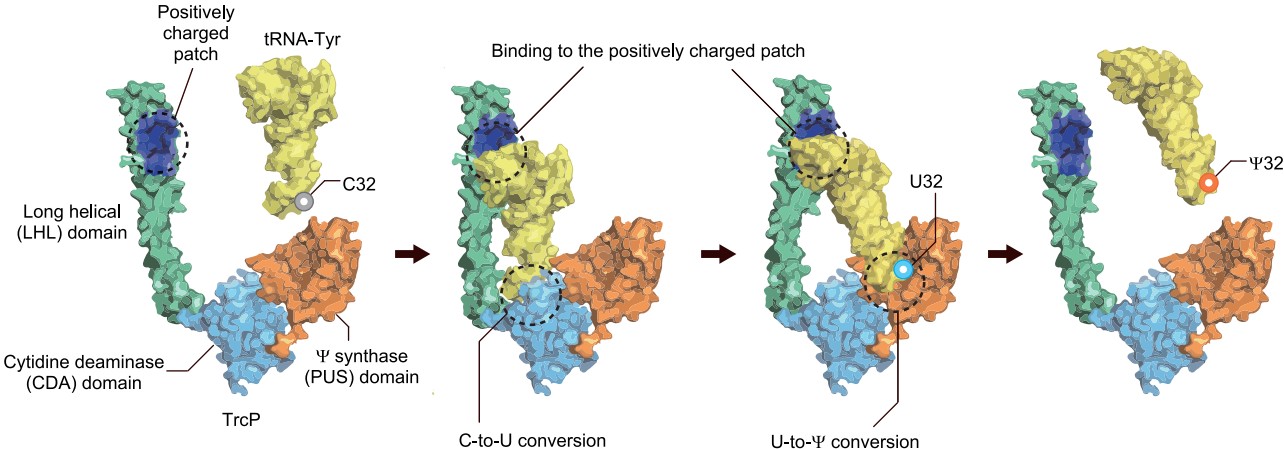

**Fig. 7 | Model of TrcP mediated C-to-Ψ editing.** The anticodon loop of tRNA-Tyr locates in the catalytic pocket of the CDA (light blue) domain and undergoes C-to-U conversion. Then the anticodon moves to the PUS domain (orange) and undergoes U-to-Ψ conversion. During both reactions, the positively charged patch (dark blue) in the LHL domain (green) supports the reaction by binding to the upper part of the tRNA. The LHL domain appears to enable the enzyme to adopt a substrate channeling mechanism to carry out its consecutive deamination and pseudouridylation reactions.

have been described[17,31], but those reactions are carried out by independent editing and modifying enzymes. TrcP catalyzes both editing and modification in two consecutive reactions (Fig. 7). First, TrcP's cytidine deaminase (CDA) domain in its N-terminus catalyzes cytidine deamination to form uridine using water as a substrate. Then, TrcP's pseudouridine synthase (PUS) domain isomerizes uridine into pseudouridine. AlphaFold-2-based structural modeling was instrumental for uncovering TrcP's structure function relationships. This machine learning algorithm predicted that TrcP contains three domains. Its two globular domains proved to correspond to its CDA and pseudouridylase. The unexpected third long helical (LHL) domain proved critical for TrcP substrate binding. This unique domain appears to enable the enzyme to adopt a substrate channeling mechanism to carry out its consecutive deamination and pseudouridylation reactions. Additional studies of the control of C-to-Ψ editing uncovered an interdependent network of modification in the tRNA anticodon loop whereby C-to-Ψ editing at position 32 requires methyl-thio modification of A at position 37 and suppresses the conversion of G to Q at position 34. Functionally, this modification circuit facilitates coupling of tRNA modification states to environmental iron availability and impacts decoding.

We propose that the positively charged patch in TrcP's LHL domain enables tRNA substrate channeling. Since this patch facilitated both C-to-U editing and pseudouridylation, this interaction likely keeps substrate tRNAs bound to the enzyme during both the deamination and isomerization reactions (Fig. 7). It is tempting to speculate that continuous binding of the upper part of substrate tRNA by the positively charged patch and flexible movement of the long helical domain facilitates the two reactions by directing the movement of the anticodon loop from the CDA to the PUS domain without diffusion. Metabolic enzymes adopt substrate channeling mechanisms for efficient consecutive reactions in which substrate molecules are conveyed from one enzyme to another without diffusion. Biosynthesis of some macromolecules, including tRNAs, depend on channeling by their associated protein factors. For example, in the biogenesis of cysteinyl- or glutaminyl-tRNAs, SepRS and SepCysS or GatCAB and GluRS form complexes catalyzing two-step aminoacylation, respectively, thereby sequestering misacylated tRNAs from the active aminoacyl-tRNA pool[41,42]. In these cases, the complexes stably bind the tRNA core during two consecutive reactions and channel the 3′ end of tRNAs from one catalytic site to the other. This mechanism is analogous to the proposed mechanism of TrcP channeling based on the tethering of tRNA substrates by the LHL domain.

Our results suggest that the TrcP-CDA domain is a previously undescribed zinc-dependent RNA deaminase domain. TrcP structurally resembles zinc-dependent deaminases (Fig. 3b and Supplementary Fig. 5), although its primary sequence is distinct from other deaminases (Supplementary Fig. 1). The conserved cysteines in TrcP essential for C-to-U editing are structurally well aligned with the cysteines in the Blasticidine-S deaminase (BSD) that coordinate a zinc ion (Fig. 3b, Supplementary Fig. 5), strongly suggesting that TrcP also utilizes zinc ion for its reaction. A notable difference between TrcP and other deaminases is in the putative catalytic glutamate residue (E303 in TrcP and E59 in Blasticidin-S deaminase, Fig. 3b and Supplementary Fig. 5). Although the spatial coordinates of the cysteine cluster and the catalytic glutamate's carboxylate group in the active sites of TrcP and BSD are similar, the orientation of the side chain and position of these glutamate residues in TrcP and BSD (E303 and E59, respectively) differ (Fig. 3b and Supplementary Fig. 5). This difference raises the possibility that these structurally similar deaminase catalytic pockets arose via distinct evolutionary pathways.

The AlphaFold-2 assisted identification of a new deaminase family suggests that additional deaminase families, which cannot be identified by in silico searches based solely on primary protein sequences without biochemical or genetic data await discovery. Such editing enzymes may account for editing reactions in plants, kinetoplastids, and marsupials, where editing enzymes remain unidentified[17]. Coupling structural prediction and structural homology search has great potential for identification of deaminases and other enzyme families.

C-to-Ψ editing is part of an interdependent network of modifications within the tRNA-Tyr anticodon loop. C-to-Ψ editing is promoted by 2-methylthio modification (ms2) in $ms^2io^6A$ at position 37 and partially suppresses queuosine (Q) formation at position 34 (Fig. 5f). Interdependencies in tRNA modifications between individual modifications have been reported, especially in the anticodon loop, but networks of more than two modifications are unusual. One example is in the thermophilic bacterium *Thermus thermophilus*, where Ψ55 modification facilitates both $m^1A58$ and $m^5s^2U54$ modifications[43]. These latter two modifications are critical for tRNA thermostability, suggesting that Ψ55 optimizes tRNA stability depending on the temperature by controlling the frequency of the two modifications. Similarly, the modification network found in *V. cholerae* tRNA-Tyr can function as a circuit that couples modification states to environmental conditions. We found that the amount of C-to-Ψ editing varies with environmental iron concentrations, presumably because iron is required for generation of the iron-sulfur cluster in MiaB[44] (Fig. 5f).

Such changes in modification states can facilitate adaptation to iron-depleted conditions, including within host niches. Environmental control of post-transcriptional editing can provide a fitness advantage that does not depend on genomic mutations in tRNA genes.

## Methods

### Strains and culture conditions

The strains used in this study are listed in Supplementary Data 3. *V. cholerae* C6706, a clinical isolate, was used as the wild-type strain[45]. All *V. cholerae* strains were grown in LB containing 1% NaCl at 37 °C. *E. coli* SM10 (lambda pir) harboring derivatives of pCVD442 was cultured in LB plus carbenicillin (Cb). Antibiotics were used at the following concentrations: 200 µg/mL streptomycin, 50 µg/mL Cb, 50 µg/ml kanamycin (Km). A transposon insertion strain (*tgt*::Tn) is derived from the strain library[46].

### Strain and plasmid construction

All mutations in C6706 were created using homologous recombination and a derivative of the suicide vector pCVD442. Targeting vectors for gene deletions contained -1000 bp of DNA flanking each side of the target gene cloned into pCVD442's SmaI site using isothermal assembly[47].

For complementation experiments, *trcP* or fragments of the *trcP* ORF were cloned into pHL100's SmaI site with the 23 nt upstream sequence and the Flag tag sequence on the C-terminal using NEBuilder HiFi DNA Assembly Master Mix (NEB); subsequently, plasmids encoding *trcP* mutants were generated with the PrimeSTAR Mutagenesis Basal Kit (TAKARA-bio).

TrcP protein expressing vector pET28-TrcP was generated by integrating *trcP* ORF and a linearized pET28 using the NEBuilder HiFi DNA Assembly Master Mix (NEB). The mutant derivatives of this vector were generated with the PrimeSTAR Mutagenesis Basal Kit (TAKARA-bio).

Reporter protein-expressing plasmids for measuring Tyr and Ile decoding ability were generated by mutagenesis of pBAD-RFlucUCG[40] encoding Renilla and Firefly luciferases using Takara PrimeSTAR (Clontech).

Plasmids used in this study are listed in Supplementary Data 4.

### RNA extraction

Total RNA was extracted with TRIzol (Life Technologies) according to the manufacturer's instructions.

### Isolation of individual tRNA species

One liter or 500 ml cultures of log-phase ($OD_{600} = 0.3–0.4$) and stationary phase (24 h) *V. cholerae* cells were harvested, and total RNA was extracted[10]. Briefly, cells were resuspended in 5 mL buffer (50 mM NaOAc, pH 5.2, 10 mM Mg(OAc)$_2$), mixed with 5 mL water-saturated phenol, and agitated vigorously for 1 h in a 50 ml Erlenmeyer flask with a stir bar. The aqueous phase was separated by centrifugation, washed with chloroform, and recovered by isopropanol precipitation. RNA was run through a manually packed DEAE column (GE healthcare) to remove contaminants and the rRNA fraction. Typically for a 1 L culture, total RNA resolved in 10 ml of the equilibration buffer (100 mM Hepes-KOH pH 7.4) and was loaded on 2 ml of DEAE beads equilibrated with 20 ml of the equilibration buffer. After sequential washes with 10 ml of the equilibration buffer and 10 ml of wash buffer (100 mM Hepes-KOH pH 7.4 and 300 mM NaCl), the RNA fraction was eluted with 10 ml of the elution buffer (100 mM Hepes-KOH pH 7.4 and 1 M NaCl) and recovered by isopropanol precipitation. Individual tRNA species were purified using biotinylated DNA probes anchored to a high-capacity streptavidin agarose resin (GE Healthcare). Typically for this purification, 2 mg of RNA treated with DEAE beads is mixed with 200 µl of beads which are bound to 4 nmol of probes in 30 mM Hepes-KOH, pH 7.0, 1.2 M NaCl, 15 mM EDTA, and 1 mM DTT at 68 °C for 30 min with shaking. Beads were washed three times with 15 mM Hepes-KOH, pH 7.0, 0.6 M NaCl, 7.5 mM EDTA, and 1 mM DTT and seven times with 0.5 mM Hepes-KOH, pH 7.0, 20 mM NaCl, 0.25 mM EDTA, and 1 mM DTT. Purified tRNAs were extracted from beads with TRIzol. After Turbo DNase (Thermo Fisher Scientific) treatment to remove residual DNA probes, tRNAs were purified on 10 % TBE-Urea gels. The probes used in this study are listed in Supplementary Data 5.

### Purification of recombinant proteins

The BL21(DE3) strain transformed with pET28b encoding WT and mutant TrcP proteins, RluA, or the chimeric TrcP-RluA protein (NTD-RluA), was grown in 10 ml LB medium (Km50) overnight and inoculated into 1 L LB medium (Km50) and grown at 37 °C with shaking. When optical density ($OD_{600}$) reached 0.3, the flask was moved to 18 °C an incubator and shaken for 30 min. Protein expression was induced by the addition of 1 mM Isopropyl β-D-1-thiogalactopyranoside (IPTG) and the flask was incubated with shaking at 18 °C for 24 h. Harvested cells were resuspended in 40 ml lysis buffer (50 mM Tris-HCl pH 8.0, 10 mM MgCl$_2$, 10% glycerol, 300 mM NaCl, 0.2 U/mL Dnase I, 1 mM PMSF, complete proteinase inhibitor mixture; Roche) and homogenized with an EmulsiFlex for 20 min. Cleared lysate (35 ml) supplemented with 700 µl of 2 M imidazole (final concentration 40 mM) was mixed with 1.5 ml Ni-NTA beads equilibrated with 10 ml lysis buffer and incubated at 4 °C for 2.5 h with gentle rotation. Protein bound beads were loaded on an open column (Biorad) and washed twice with 10 ml wash buffer (50 mM Tris-HCl pH 8.0, 10 mM MgCl$_2$, 10% glycerol, 300 mM NaCl, 40 mM imidazole). Protein was eluted with Elution buffer 1 (50 mM Tris-HCl pH 8.0, 10 mM MgCl$_2$, 10% glycerol, 300 mM NaCl, 250 mM imidazole) and Elution buffer 2 (50 mM Tris-HCl pH 8.0, 10 mM MgCl$_2$, 10% glycerol, 300 mM NaCl, 400 mM imidazole). The two elution fractions were mixed and dialyzed overnight in Dialysis buffer 1 (20 mM Tris-HCl pH 8.0, 300 mM NaCl, 10% Glycerol, 1 mM DTT) and 8 h in Dialysis buffer 2 (20 mM Tris-HCl pH 8.0, 150 mM NaCl, 10% Glycerol, 1 mM DTT). Protein concentration was measured by Qubit (Invitrogen). For measuring melting temperatures, WT and KR mutant TrcP were purified with ion-exchange chromatography using a 1 ml HiTrap Heparin HP column (Cytiva) and eluting across a 150–1000 mM NaCl gradient. Target protein fractions were pooled and concentrated with Amicon columns (Millipore-Sigma).

For measurement of zinc concentration, highly purified, untagged TrcP was prepared using a previously described SUMO2-based protein purification strategy[48]. Briefly, *V. cholerae* TrcP was sub-cloned into a custom pET vector and expressed as a 6× His-tagged N-terminal human SUMO2 fusion in *E. coli* BL21 RIL bacteria (Agilent). A 50 ml starter culture grown overnight at 37 °C in MDG medium (1.5% Bacto agar, 0.5% glucose, 25 mM Na$_2$HPO$_4$, 25 mM KH$_2$PO$_4$, 50 mM NH$_4$Cl, 5 mM Na$_2$SO$_4$, 0.25% aspartic acid, 2–50 µM trace metals, 100 µg mL$^{-1}$ ampicillin, 34 µg mL$^{-1}$ chloramphenicol) was used to seed growth of 2 × 1 L induction cultures grown in M9ZB medium (47.8 mM Na$_2$HPO$_4$, 22 mM KH$_2$PO$_4$, 18.7 mM NH$_4$Cl, 85.6 mM NaCl, 1% Cas-Amino acids, 0.5% glycerol, 2 mM MgSO$_4$, 2–50 µM trace metals, 100 µg mL$^{-1}$ ampicillin, 34 µg mL$^{-1}$ chloramphenicol). M9ZB cultures were grown at 37 °C with shaking at 230 rpm until an optical density at 600 nm ($OD_{600}$ nm) of -2.5. Cultures were then transferred to an ice bath for 20 min, supplemented with 0.5 mM IPTG and incubated at 16 °C with shaking at 230 rpm for overnight growth. Harvested cell pellet was lysed by sonication in lysis buffer (20 mM Hepes-KOH pH 7.5, 400 mM NaCl, 30 mM imidazole, 10% glycerol and 1 mM DTT) and recombinant TrcP was purified from clarified lysates using Ni-NTA resin (Qiagen) and gravity chromatography. Ni-NTA resin was washed with lysis buffer adjusted to 1 M NaCl concentration and eluted with lysis buffer adjusted to 300 mM imidazole concentration. Purified TrcP was then supplemented with -250 µg of human SENP2 protease (D364–L589, M497A) and dialyzed overnight at 4 °C in buffer (20 mM HEPES-KOH pH 7.5, 200 mM NaCl and 1 mM DTT). TrcP was next purified by ion-

exchange chromatography by binding to a 5 ml HiTrap Heparin HP column (Cytiva) and eluting across a 150–1000 mM NaCl gradient. Target protein fractions were pooled and further purified by size-exclusion chromatography using a 16/600 S75 column (Cytiva) with a running buffer of 20 mM Hepes-KOH pH 7.5, 250 mM KCl and 1 mM TCEP-KOH. Final protein was concentrated to ~6.3 mg/mL, flash-frozen in liquid nitrogen, and stored at −80 °C

### Nucleoside analysis

Purified and sometimes derivatized tRNA-Tyr (100 ng – 1 µg) was digested with 0.5 unit Nuclease P1 and 0.1 unit of phosphodiesterase I in 22 µl reactions containing 50 mM Tris-HCl pH 5.3, 10 mM $ZnCl_2$ at 37 °C for 1 h. Then, reaction mixtures were mixed with 2 µl 1 M Tris-HCl pH 8.3 and 1 µl of 1 unit/µl Calf Intestine phosphatase and incubated at 37 °C for 30 min. Enzymes were removed by filtration using 10 K ultrafiltration columns (VWR) and 5–10 µl of digests were injected into an Agilent 1290 uHPLC system bearing a Synergi Fusion-RP column (100 × 2 mm, 2.5 µm, Phenomenex) at 35 °C with a flow rate of 0.35 ml/min and a solvent system consisting of 5 mM $NH_4Oac$ (Buffer A) and 100 % Acetonitrile (Buffer B). The gradient of acetonitrile was as follows: 0 %; 0–1 min, 0–10 %; 1–10 min, 10–40 %; 10–14 min, 40–80 %; 14–15 min, 80–100 %; 15–15.1 min, 100 %; 15.1–18 min, 100–0 %; 18–20 min, 0 %; 20–26 min. The eluent was ionized by an ESI source and directly injected into an Agilent 6460 QQQ. The voltages and source gas parameters were as follows: gas temperature; 250 °C, gas flow; 11 L/min, nebulizer; 20 psi, sheath gas temperature; 300 °C, sheath gas flow; 12 L/min, capillary voltage; 1800 V, and nozzle voltage; 2000 V. Dynamic multiple reaction monitoring (MRM) was carried out to detect labeled and unlabeled known modifications. The retention time windows and $m/z$ values of precursor and product ions for dynamic MRM analyses are listed in Supplementary Data 6.

### Direct sequencing

cDNAs were synthesized from 10 ng purified tRNA-Tyr using Super Script III (Invitrogen) using the RT primer. cDNA was amplified by PCR using primers: RT-PCR primers. PCR products were sequenced by Sanger sequencing using the sequencing primer. The sequences of the primers are listed in Supplementary Data 5.

### Oligo protection

In total, 50 pmol of tRNA-Tyr was mixed with 500 pmol of two DNA oligos complementary to C32 and U32 tRNA-Tyr (Supplementary Data 5) in 50 µl aliquots containing 50 mM Hepes-KOH pH 7.6, 150 mM KCl and heated to 90 °C for 1 min and gradually cooled down to room temperature at 1 °C/min for annealing, followed by RNase digestion with 50 ng RNase A and 50 unit RNase $T_1$ on ice for 15 min. Protected DNA/RNA duplexes were purified on 10 % TBE-Urea gels and recovered by isopropanol precipitation and dissolved in 5 µl milliQ water. A 2 µl aliquot was subjected to fragment analysis with RNase A using a MALDI-TOF spectrometer as described above.

### Cyanoethylation

Cyanoethylation was done as described[19]. 4 µl of RNA (1–2 µg) was mixed with 30 µl CE solution (50% ethanol, 1.1 M triethylamine pH 8.6) and 4 µl of 15.2 M acrylonitrile, followed by incubation at 70 °C for 1 h. The reaction was stopped by addition of 162 µl milliQ water and by placing the reaction tube on ice. RNA was recovered by ethanol precipitation.

### Fragment analysis

In total, 400–1000 ng partial fragments of tRNA-Tyr were digested in 3 µl aliquot with 20 ng RNase A (QIAGEN) in 10 mM $NH_4Oac$ pH 7, or 20 unit RNase $T_1$ in 10 mM $NH_4Oac$ pH 5.3 at 37 °C for 1 hr. On a MALDI steel plate, 0.5 µl of matrix (0.7 M 3-hydroxypicolinic acid (HPA) and 70 mM ammonium citrate in 50 % milliQ water and 50 % acetonitrile) was mounted and dried, followed by mounting of 0.5 µl Rnase digests and drying. The samples were analyzed with Bruker Ultraflex Xtreme MALDI-TOF mass spectrometer.

### In vitro editing reaction

Cytidine deamination activity of TrcP was measured by direct Sanger sequencing of cDNA derived from tRNA-Tyr in Figs. 1b, 4d and 5c. tRNA-Tyr (5 pmol) was incubated with 0–15 pmol TrcP recombinant protein in 20 µl reaction mixtures (50 mM Tris-HCl pH 8.0, 5 mM $MgCl_2$, 30 mM NaCl) for 1 hr at 37 °C. The reaction was stopped by an addition of 100 µl phenol chloroform and vigorous vortexing; subsequently, tRNAs were washed with chloroform and recovered by iso-propanol precipitation.

C-to-U editing and pseudouridylation activity was measured by MALDI-TOF-MS in Figs. 1c, 2c and 5a. tRNA-Tyr isolated from cells or in vitro transcription reactions (50 pmol) was incubated with 0, 5, or 50 pmol TrcP and TrcP mutant proteins in 200 µl reaction mixture (50 mM Tris-HCl pH 8.0, 5 mM $MgCl_2$, 30 mM NaCl) for 1 hr at 37 °C, and tRNAs were recovered as described above. The partial fragment of tRNA-Tyr containing position 32 was purified by oligo protection, and pseudouridines were labeled by cyanoethylation. C, U, and Ψ containing fragments were measured by fragment analysis by MALDI-TOF-MS as described above.

Stable isotope labeled water ($H_2^{18}O$) (Sigma Aldrich) was used for identifying the oxygen source of deamination (Fig. 3f) and tracking pseudouridylation by nucleoside analysis (Fig. 4f). In Fig. 3f, 5 pmol tRNA-Tyr isolated from the ΔtrcP mutant was incubated for 1 hr at 37 °C with 10 pmol TrcP protein in a 20 µl reaction in (50 mM Tris-HCl pH 8.0, 5 mM $MgCl_2$ 30 mM NaCl) containing 10 µl $^{18}O$-labeled water. tRNAs were recovered as described above and signals of natural and heavier pseudouridines were detected by nucleoside analysis as above. In Fig. 4f, 200 pmol tRNA was incubated with 200 pmol TrcP-NTD in a 40 µl reaction in (50 mM Tris-HCl pH 8.0, 5 mM $MgCl_2$, 60 mM NaCl) containing 20 µl $^{18}O$-labeled water. tRNAs were recovered and subject to reaction with WT and mutant TrcP proteins, RluA, and chimeric RluA proteins. 5 pmol tRNA was incubated with 5 pmol or 500 fmol proteins in a 20 µl reaction in (50 mM Tris-HCl pH 8.0, 5 mM $MgCl_2$, 75 mM NaCl) for 1 hr at 37 °C. tRNAs were recovered and subject to nucleoside analysis as described above and signals of U + 2 normalized with $m^5U$ signals were tracked as the readout of uridine to pseudouridine conversion.

### Gel mobility shift assay

The substrate tRNA binding capacities of TrcP and the KRmutant were evaluated using a gel mobility shift assay. Isolated tRNA-Tyr from the ΔtrcP strain and tRNA-Asp from the WT strain were labeled with $^{32}P$ at their respective 5′ ends. Briefly, calf intestinal alkaline phosphatase (CIP) (NEB) was used to dephosphorylate tRNAs, which were purified with gel electrophoresis, and labeled with $[\gamma-^{32}P]ATP$ by T4 PNK (NEB), followed by gel purification. WT or KRmut TrcP proteins (0–5 µM) were incubated with 1 µM cold tRNA-Tyr or tRNA-Asp and a trace amount of $^{32}P$-labeled tRNAs (<0.1 µM) in a 5 µl reaction in 5 mM $MgCl_2$, 50 mM Tris-HCl pH 8.0, and 60 mM NaCl at 37 °C for 10 min, followed by mixing with loading dye (80% glycerol and 0.05% bromophenol blue) and gel electrophoresis for 30 min at 180 V and 4 °C on 4% polyacrylamide gel containing 50 mM Tris-Acetic acid pH 8.0 and 5 mM $Mg(Oac)_2$; the running buffer was the same composition as the gel. Gels were dried with a gel dryer (Biorad) and bands were detected using an FLA-5000 phosphoimager (Fuji Film).

### Translation reporter assay

The WT, ΔtrcP, Δtgt, and ΔtrcP/Δtgt strains were transformed with the frameshift or 0-frame reporters. Four to six colonies were picked, and cultured overnight in 1 ml LB medium (Cb 50 µg/ml) at 37 °C and then

diluted 500-fold in 200 µl LB medium (Cb 50 µg/ml) and cultured at 37 °C. For log phase cultures, 2 µl of 20% arabinose was added to each culture and harvested when the OD600 reached 0.2–0.4. For stationary phase cultures, 2 µl of 20% arabinose was added one hour after the $OD_{600}$ reached 0.2–0.4 and harvested after another one hour. The cells were spun down and resuspended in 100 µl RPMI1640 medium (Gibco) supplemented with 400 µg/ml lysozyme (Sigma Aldrich). The cell suspension was incubated on ice for 10 min, flash-frozen with liquid nitrogen, and thawed on iced water. Luciferase activities were measured with Dual-Glo luciferase assay system (Promega) by SpectraMax i3x (Molecular Devices). The lysate (20 µl) was mixed with 20 µl Dual-Glo luciferase assay reagent and incubated at room temperature for 10 min, followed by measurement of Firefly luminescence on a plate reader. Dual-Glo Stop & Glo reagent (20 µl) was mixed with the reaction mixtures, and incubated at room temperature for 10 min, followed by the measurement of Renilla luminescence. F/R values (the ratio of Firefly to Renilla signals) derived from 0 frame and +1 frame constructs were calculated as Rt and Rs, representing the efficiency of in-frame decoding and +1 frameshifting, respectively. Then, Rt/Rs values were calculated to estimate the Tyr decoding efficiency. Rt/Rs values were normalized with the average value of WT strain and shown as relative Rt/Rs values. Each data point represents the result derived from an independent culture.

### Measuring zinc ion concentration

The zinc concentration of TrcP in solution was measured with a Zinc Assay Kit (Abcam) according to the manufacturer's instruction. Twenty-five µl of TrcP solution (97.6 µM) was mixed with 25 µl 14% trichloroacetic acid or water, followed by centrifugation and recovery of the supernatant. A 50 µl serial dilution series of a Zn standard solution (0–100 µM) and TrcP samples were mixed with a 200 µl reaction mix and incubated for 10 min at room temperature. Light absorption at $A_{560}$ was measured. Zinc concentrations of TrcP solutions were calculated with a standard curve generated with the standard Zn solutions.

### Complementation assay

A *V. cholerae* Δ*trcP* strain was transformed with plasmids expressing TrcP or mutant derivatives. The resultant strains were cultured in LB medium (Km 50 µg/ml) overnight at 30 °C. Overnight cultures were diluted to $OD_{600}$ 0.01 in 5 ml LB (Km 50 µg/ml, IPTG 10 µM) and cultured at 37 °C until $OD_{600}$ reached 0.3. Culture (1 ml) was transferred to two tubes, and cells were harvested by centrifugation. One tube was used to assess RNA editing, and the other was used for measuring TrcP and mutant protein expression levels. RNA editing frequency was measured by Sanger sequencing as described above. TrcP and mutant protein expression levels were measured by western blotting. For the latter assays, cell pellets were resuspended in 200 µl lysis buffer (50 mM Tris-HCl pH 8.0, 10% glycerol, and 300 mM NaCl) and homogenized by sonication, and 10 µl of lysate mixed with loading solution and boiled and loaded on NuPAGE 4–12% Bis-Tris Gel and electrophoresis was carried out for 40 min at 200 V. An iBlot2 (Invitrogen), was used to transfer proteins nitrocellulose membranes, followed by blocking with TBST buffer (20 mM Tris-HCl pH 7.6, NaCl 150 mM, 0.1% Tween-20) supplemented with 5% milk, primary (Sigma anti-Flag M2 antibody cat#F3165; 1 in 1000 dilution for TrcP and its mutants, Santa Cruz RpoB antibody cat#sc-56766,8RB13; 1 in 2000 dilution for RpoB) and secondary antibody (Invitrogen anti-mouse antibody cat#31430, 1 in 2000 dilution) binding, signals were detected with Supersignal West Pico Plus Chemiluminescent substrate (Thermo Fisher) using a Chemidoc (Biorad).

### Structural modeling

The amino acid sequences of full-length TrcP and its CDA domain (1–114 aa and 250–339 aa) were input to ColabFold2 program[28]. The first model among five nearly identical models was used for further analyses. A pdb file was generated and used for structural homology searches using the Dali Server[30]. The resultant structures were overlaid on the structural model of the CDA domain (Fig. 3b and Supplementary Fig. 5). Electro potential maps were generated by APBS Electrostatics in PyMOL (Schrödinger) with a default setting.

### Phylogenetic analysis of TrcP

TrcP-NTD (1–339 aa) homologs were searched using BLAST with default algorithm parameters except for Max target sequences (increased to 20000), yielding 4862 hits, including 4567 sequences derived from 959 species with E-value less than 1 e-20 (Supplementary Data 7). To depict the distribution of TrcP-NTD homologs, a phylogenetic tree was generated by phyloT[49] with manually picked 757 organisms, including 38 organisms having TrcP homologs with E-value less than 1 e-20, and displayed by iTOL[50] (Supplementary Data 8 and Supplementary Fig. 3). Sequence alignment was performed with Cluxtal X2[51] and displayed with Bioedit[52].

### Thermofluor assay

Thermofluor assays for melting temperatures were conducted as described[53,54]. Proteins purified on heparin columns and concentrated with Amicon Ultra Centrifugal Filter units (10 Kda) were assayed. Aliquots (20 µl) consisting of 20 mM Hepes pH 7.5, 250 mM NaCl, 1 mM TCEP-KOH, 340 nM WT or KR mutant TrcP, were mixed with 2 µl 10 × SYPRO Orange dye (Thermo Fisher) and loaded in a StepOne Real-Time PCR System (Applied Biosystems). The temperature was raised from 20 °C to 95 °C, and measurements were made at a 0.3 °C interval (+0.63 °C /min). The first derivative lines were calculated and depicted by GraphPad Prism. The average lines from the three reactions are shown.

### Reporting summary

Further information on research design is available in the Nature Research Reporting Summary linked to this article.

## Data availability

Data is available upon requests. PDB entries used in this study: 3OJ6 [https://doi.org/10.2210/pdb3oj6/pdb] (blasticidine-S deaminase), 4EG2 [https://doi.org/10.2210/pdb4eg2/pdb] (cytidine deaminase). The coordinates for the structural models of TrcP and its derivatives are provided in Supplementary Data 9–11. Source data are provided with this paper.

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

## Acknowledgements

We thank members of the Waldor lab and Vlad Denic for comments on the manuscript and Sadie Antine for technical assistance and the Harvard FAS Science Core Facility for use of the MALDI equipment. This study was supported by AI-042347 and HHMI to M.K.W., and Pew Biomedical Scholars program to P.J.K..

## Author contributions

S.K. and M.K.W. designed research; S.K. and V.S. performed experiments; S.K analyzed data; S.K. and K.L.M. purified proteins; S.K., V.S., M.K.W., P.C.D., and P.J.K. discussed the results; S.K. and M.K.W. wrote the paper.

## Competing interests

The authors declare no competing interests.
