## [Peer Review File · Nature Communications]

Title: Sequential action of a tRNA base editor in conversion of cytidine to pseudouridineEditorial Note: Parts of this Peer Review File have been redacted as indicated to remove third-party material where no permission to publish could be obtained.

REVIEWER COMMENTS

Reviewer #1 (Remarks to the Author):

Here the authors report the characterization of a protein from *Vibrio cholera* responsible for the conversion of C32 in the anticodon loop of tRNA^{Tyr} to pseudouridine. This process requires two enzymatic steps: 1) deamination of C32 to U32 by a cytidine deaminase (CDA) and 2) isomerization of U32 to pseudoU32 by a pseudouridine synthetase (PUS). Previous studies established that a single gene (trcP) was required for this conversion. Here the authors carry out extensive biochemical characterization of the TrcP protein establishing the domains responsible for the individual steps of the process, including the domain responsible for cytidine deamination that contains a previously uncharacterized long helical domain (LHD) insertion. The overall architecture of this RNA editing enzyme, with its CDA and PUS domains along with the LHD for tRNA recognition, is highly unique. Furthermore, the discovery of a single protein capable of both cytidine deamination and pseudouridylation, along with the proposed channeling mechanism mediated by the positively charged patch on the LHD, is ground breaking. This is an important paper for the RNA editing field. The paper is well written and the conclusions are largely supported by the results. I have the following comments.

- 1) When referring to Figure 1B, the authors indicate that the concentration of TrcP is shown when the quantity of the enzyme in picomoles is given.
- 2) In Figure 1C, 2C, and 5A, mass spectra are shown for the conversion of nucleotides 29-32 of the tyrosine tRNA substrate from a mass of 1325 to a mass of 1379 (for deamination, pseudouridylation and cyanoethylation). Yet in each case, a prominent peak is also observed at approximately 1344 but no comment or explanation is offered. In Figure 2C, it appears this species is more abundant with TrcP and NTD treatments but less so with the CTD. Can the authors elaborate on the possible origin of this mass?
- 3) When testing for the importance of the LHD positive patch, a mutant is described that has six positively charged residues all mutated to alanine at the same time. Given that such a drastic change in protein sequence could cause a misfolding resulting in loss of activity, the authors should test single point mutants or have another test for folding of the 6X mutant (such as circular dichroism of the purified protein).
- 4) Similarly, for tests of mutations at putative metal binding residues, the authors should test mutants that would be expected to retain some metal binding (e.g. H, D, E or S) to see if reduced activity is observed (since the alanine mutants were all completely lacking activity).

5) The experiments using dipyriddyI to evaluate the effect of iron chelation on C32 editing is lacking a control showing the effect is not a general one but specific to this editing system. The authors should monitor other editing or modification sites in this or other tRNAs after dipyriddyI treatment to demonstrate the effect is specific to this site.

Reviewer #2 (Remarks to the Author):

Despite the numerous RNA modifications that have been described, few cases of forcibly sequential modification reactions have been described to date. For example, the formation of wybutosine from m1G, the glycosylation of Q, C to m3U, and A to m1I, etc. The present manuscript by Kimura et al. Shows that at position 32 of tRNATyr in *Vibrio*, an encoded C is first edited to U followed by conversion of the edited U to pseudouridine. Notably, they identify a protein involved in these reactions, which is a modular protein with 3 main domains: 1) an RNA binding domain, 2) a deaminase domain, and 3) a pseudouridine synthase domain. This is supported by biochemical assays and careful analysis by sequencing and mass spectrometry. Furthermore, the influence of other anticodon modifications is suggested, and some data presented for it, as well as the possible role of the C32 editing and modification in translation. Overall, this is a very interesting story, and the observations are important in expanding the numbers and types of sequential modifications. However, there are several issues that need to be addressed, some major.

Major comments:

1. In figure 1, it is not clear if they directly analyzed the AGACp oligo or was the total nucleoside analysis done in the context of the larger (nt 10-46) fragment? Along these lines, the authors should provide a gel with the results of the biotin-oligo purification. Without a notion of purity, how can we be sure that the C to U and the U to pseudoU are happening at the same nucleotide position, instead of two different positions. Did the authors sequence the tRNA fragment by mass spectrometry to connect their cDNA sequencing result to the actual RNA oligo? All this needs either additional data or major clarification.
2. If the authors did total nucleoside analysis of the purified tRNATyr, they should provide the LC-MS data showing that only the modifications shown in the tRNATyr cartoon (Fig. 1) are found in their purified tRNA. This will in fact prove its purity.
3. In the binding assay. It is not clear, why the data goes from little binding to a lot more binding when comparing 2 μ M to 5 μ M protein (WT). Also, the amount of tRNA used is a formula for non-specific binding. For the binding to be meaningful and to rule out non-specific binding mediated by the putative RNA-binding domain, the authors should at least provide an apparent dissociation constant with proper quantification.
4. When making the claim that tRNA without Q or for that matter ms2io6A influence pseudouridine formation etc. The authors should provide evidence that the tRNAs purified from those mutants indeed lack those modifications. It has been my experience that several so-called deletion mutants are not quite mutated when tested carefully.

5. In the dual reporter assays, an important control is missing. They should provide a similar control for reading U and C-ending codons for tRNAs that do not undergo C to pseudouridine, if the claim is that this effect is specific for the editing and modification in question.
6. The authors should explain why no data is presented on the use of a transcript to test the activities described. If this was mentioned on the paper, I have missed and should therefore discuss/emphasized.

Reviewer #3 (Remarks to the Author):

In this manuscript by Kimura et al., the authors test the enzymatic activity of a putative tRNA modification enzyme in *Vibrio cholerae*. Through a series of biochemical, molecular and genetic experiments, the authors show that tRNA C-to-Psi conversion (TrcP) protein catalyzes both cytosine deamination and pseudouridylation. The authors then use mutagenesis and truncation analysis to demonstrate that the deaminase domain alone is necessary and sufficient for deamination while the pseudoridylase domain requires linkage to the deaminase domain for activity. Moreover, the paper finds that TrcP activity is stimulated by another modification at position 37 of the tRNA while TrcP-dependent pseudouridylation at position 32 affects the levels of queuosine modification at position 34. Finally, experiments using frameshifting reporter assays suggest that pseudouridine modification impacts decoding of UAU codons in coordination with the queuosine modification. The experiments and assays are well-designed and executed with appropriate controls and standards.

Overall, the results from this paper demonstrate conclusively that TrcP is a tRNA modification enzyme with two distinct activities. These results are noteworthy because they provide insight into a novel molecular mechanism by which cytosine is converted to pseudouridine in tRNA. Since this is the first instance of a single enzyme that catalyzes both editing and pseudouridylation, these findings will be impactful for the RNA field and for future identification of similar enzymes. Based upon the insight provided by these findings, this manuscript is recommended for acceptance with only minor questions and edits noted below.

Points

- The use of “editing” when referring to tRNA modifications can be tricky. Due to historical reasons, some in the field might reserve the term “editing” for only C to U and A to I modifications in mRNA. One suggestion to avoid this potential issue is to replace “editing” with “modification”. Instead of referring to the TrcP activity as “editing”, the Authors could refer to cytosine to uridine conversion as deaminase activity since that is the specific biochemical activity catalyzed by TrcP. Moreover, the title could read “Sequential action of a tRNA modification enzyme in conversion of cytidine to pseudouridine”.
- Page 3, line 54. The authors state that “However, with some exceptions, the functions and biosynthesis pathways of editing in tRNAs are largely unclear”. This statement makes it seem that very little is known about the proteins that catalyze C to U or A to I conversion in tRNAs. However, the

biosynthetic pathways for tRNA editing are known for several organisms. We suggest that the authors soften the wording in this sentence.

- In Figure 4, the TrcP mutant in the long helical domain consists of 6 mutations from positively charged residues to alanine. The resulting KRmut protein is unable to rescue modification in cells nor bind tRNA-Tyr. However, the mutation of 6 residues could have rather drastic consequences on protein folding that are not directly linked to tRNA binding. Did the authors test single point mutations in the long helical domain and did they have any influence on activity or tRNA binding? If so, we recommend that the authors include this information, even if the single point mutations had very little effect on activity or RNA binding.
- It is intriguing that the NTD containing the cytidine deaminase domain can be fused with a different pseudouridylase to yield an active enzyme in vitro. Did the authors test whether this NTD-RluA fusion protein can rescue modification in vivo?
- The experimental setup for the translation frameshift assay is unclear. In the Methods, it says “The WT, trcP, tgt, and trcP/ tgt strains were transformed with the frameshift and 0-frame reporters.” Does this mean that each strain was transformed with both the frameshift and 0-frame reporters and then fluorescence was measured? Or was each strain transformed with either the frameshift or 0-frame reporters and then each strain containing an individual reporter plasmid was measured? Does each point for each strain represent an independent culture that was measured by FACs or does each point represent technical replicates where the same sample is being measured three times? Please clarify and provide additional details in the manuscript.

REVIEWER COMMENTS

Reviewer #1 (Remarks to the Author):

Here the authors report the characterization of a protein from *Vibrio cholera* responsible for the conversion of C32 in the anticodon loop of tRNA^{Tyr} to pseudouridine. This process requires two enzymatic steps: 1) deamination of C32 to U32 by a cytidine deaminase (CDA) and 2) isomerization of U32 to pseudoU32 by a pseudouridine synthetase (PUS). Previous studies established that a single gene (*trcP*) was required for this conversion. Here the authors carry out extensive biochemical characterization of the TrcP protein establishing the domains responsible for the individual steps of the process, including the domain responsible for cytidine deamination that contains a previously uncharacterized long helical domain (LHD) insertion. The overall architecture of this RNA editing enzyme, with its CDA and PUS domains along with the LHD for tRNA recognition, is highly unique. Furthermore, the discovery of a single protein capable of both cytidine deamination and pseudouridylation, along with the proposed channeling mechanism mediated by the positively charged patch on the LHD, is ground breaking. This is an important paper for the RNA editing field. The paper is well written and the conclusions are largely supported by the results. I have the following comments.

We thank the reviewer for the careful read of our manuscript and the positive remarks.

1) When referring to Figure 1B, the authors indicate that the concentration of TrcP is shown when the quantity of the enzyme in picomoles is given.

Thank you for noting this inconsistency; we now show protein concentrations in figure 1B.

2) In Figure 1C, 2C, and 5A, mass spectra are shown for the conversion of nucleotides 29-32 of the tyrosine tRNA substrate from a mass of 1325 to a mass of 1379 (for deamination, pseudouridylation and cyanoethylation). Yet in each case, a prominent peak is also observed at approximately 1344 but no comment or explanation is offered. In Figure 2C, it appears this species is more abundant with TrcP and NTD treatments but less so with the CTD. Can the authors elaborate on the possible origin of this mass?

Thank you for the careful inspection of the figures. This fragment is derived from the tRNA D-arm (GAGDp m/z 1344). We added an explanation about the origin of this fragment in the figure legend.

3) When testing for the importance of the LHD positive patch, a mutant is described that has six positively charged residues all mutated to alanine at the same time. Given that such a drastic change in protein sequence could cause a misfolding resulting in loss of activity, the authors should test single point mutants or have another test for folding of the 6X mutant (such as circular dichroism of the purified protein).

The reviewer raises a valid point. We carried out both of the reviewer's suggestions and assessed the folding of the 6X mutant and generated single point mutants, to rule out the possibility that results with the 6X mutant were misleading due to its misfolding. The folding of the KR (6X) mutant was assessed using a Thermofluor assay^{1,2}, where melting temperatures (T_m) were determined by the peak position of first derivative curves. In this assay, the T_m of the KR mutant and the WT protein were both 35 °C (Fig. R1), suggesting that the folding of both proteins is similar. Furthermore, in the cell, the expression levels of the KR mutant and the WT were similar (Supplementary Fig.6), suggesting that there is no prominent degradation triggered in vivo by unstable protein structures. In addition, we assessed the editing activity of single alanine substitution mutants in charged residues in the LHD positive patch using a complementation assay. We generated and tested four new point mutants: R168A, R175A, K201A, K208A. Notably, each of these single alanine substitution mutants had a

reduced capacity for editing compared to the WT TrcP (Fig. R2), suggesting that individual positively charged residues in the LHD promote TrcP editing activity. Together, these distinct approaches, support the idea that the editing defect of the 6X mutant does not stem from misfolding and re-enforce the idea that the LHD positive patch facilitates editing.

Fig. R1 The first derivative of melting curves of WT and KRmutant proteins. Sypro orange dye was mixed with proteins, and the fluorescence signals were measured using the filter for FAM signals as the temperature was raised. The first derivative curves were calculated by Graphpad Prism (2nd order smoothing neighboring 8 points), and the average lines of three reactions are depicted. The melting temperatures represented by the peaks of the lines are very similar (35°C) between WT and KRmutant. This data is added to the manuscript as a new Supplementary Fig. 8.

Fig. R2 Complementation assay with single Ala substitution mutants in the positively charged patch in the LHD. The $\Delta trcP$ strain was transformed with the plasmids containing an alanine substitution mutation in the positively charged patch, and C-to- Ψ editing frequencies in the cell were measured by Sanger sequencing. All mutants showed only marginal editing activities. This data is added to the manuscript as a new Supplementary Fig. 7.

4) Similarly, for tests of mutations at putative metal binding residues, the authors should test mutants that would be expected to retain some metal binding (e.g. H, D, E or S) to see if reduced activity is observed (since the alanine mutants were all completely lacking activity).

We tested whether serine substitutions for cysteine residues in the metal binding pocket preserve some editing activity. Three new substitution mutants, where conserved cysteine residues were mutated into serine, were tested for editing activity. None of these mutants had detectable editing activity (Fig. R3), suggesting that serine cannot be substituted for cysteine residues. Thus, although serine can have metal binding activity in other proteins, in TrcP, serine does not retain zinc ion stably, or the zinc ion is not correctly coordinated in the catalytic site.

Fig. R3 Complementation assay with TrcP mutants each of which has mutations from cysteine to serine. The $\Delta trcP$ strain was transformed with the plasmids containing an serine substitution mutation in zinc ion binding pocket, and C-to- Ψ editing frequencies in the cell were measured by Sanger sequencing. All mutants showed no editing activities.

5) The experiments using dipyriddy to evaluate the effect of iron chelation on C32 editing is lacking a control showing the effect is not a general one but specific to this editing system. The authors should monitor other editing or modification sites in this or other tRNAs after dipyriddy treatment to demonstrate the effect is specific to this site.

The review raises a valid point. To assess whether iron depletion generally affects RNA editing activities, we monitored tRNA-Arg2C, where the wobble position undergoes A-to-I editing by TadA, an adenosine deaminase that belongs to the ADAT family. Sanger sequencing demonstrated that adding dipyriddy (Dip) did not change editing frequency (Fig. R4), suggesting that iron depletion does not affect editing activities in general.

Fig. R4 Dip treatment did not affect A-to-I editing efficiency at position 34 in tRNA-Arg2C. A-to-I editing occurs at position 34 in tRNA-Arg2C, and Inosine is read as "G" in Sanger sequencing. WT strains were cultured in the presence of an iron chelator 2',2'-dipyriddy (Dip), and the editing at position 34 in tRNA-Arg2C was measured by Sanger sequencing. The concentration of Dip is indicated on the left. Under all conditions, position 34 is fully edited to inosine (I)

Reviewer #2 (Remarks to the Author):

Despite the numerous RNA modifications that have been described, few cases of forcibly sequential modification reactions have been described to date. For example, the formation of wybutosine from m1G, the glycosylation of Q, C to m3U, and A to m1I, etc. The present manuscript by Kimura et al. Shows that at position 32 of tRNA^{Tyr} in *Vibrio*, an encoded C is first edited to U followed by conversion of the edited U to pseudouridine. Notably, they identify a protein involved in these reactions, which is a modular protein with 3 main domains: 1) an RNA binding domain, 2) a deaminase domain, and 3) a pseudouridine synthase domain. This is supported by biochemical assays and careful analysis by sequencing and mass spectrometry. Furthermore, the influence of other anticodon modifications is suggested, and some data presented for it, as well as the possible role of the C32 editing and modification in translation. Overall, this is a very interesting story, and the observations are important in expanding the numbers and types of sequential modifications. However, there are several issues that need to be address, some major.

We thank the reviewer for the positive comments.

Major comments:

1. In figure 1, it is not clear if they directly analyzed the AGACp oligo or was the total nucleoside analysis done in the context of the larger (nt 10-46) fragment?

As mentioned in the main text (p5, l101-l107), a fragment of tRNA-Tyr (position 10-46) was first purified and the purified fragment was digested with RNase A. Then, the mixture of RNA fragments was subjected to MALDI-TOF mass spectrometry, and tetramer oligos containing C, U, or Ψ at position 32 were detected.

Along these lines, the authors should provide a gel with the results of the biotin-oligo purification.

We only saw faint bands of rRNAs and other tRNAs, such as class I tRNAs, when we analyzed the purified RNA with gel electrophoresis. A gel purification step further removed all other RNAs (Fig. R4). Nucleoside analysis in a previous work³ confirmed that it contains only a trace amount of modifications derived from other tRNAs (see the next point as well).

Fig. R4 Purity of tRNA-Tyr after biotin-oligo purification assessed on 10% Urea-TBE gels staining with Toluidine blue O (left). The tRNA-Tyr bands were cut out from the gel and eluted.

Without a notion of purity, how can we be sure that the C to U and the U to pseudoU are happening at the same nucleotide position, instead of two different positions. Did the authors sequenced the tRNA fragment by mass

spectrometry to connect their cDNA sequencing result to the actual RNA oligo? All this needs either additional data or major clarification.

The reviewer is correct. Confirming that C-to-U and U-to- Ψ conversion happen at the same position is critical. In our previous work³, we conducted a stable isotope (SI) labeling experiment, in which SI-labeled cytidine was supplemented in the culture medium to label cytidines but not uridines. This SI-label (2 Dalton increase in mass) was observed in pseudouridine in tRNA-Tyr, indicating that labeled cytidine was incorporated into tRNA by transcription, and then labeled cytidine was converted into pseudouridine post-transcriptionally. Fragment analyses demonstrated that the SI-label was observed in the fragment including position 32, proving that C-to- Ψ conversion happens at position 32. Based on these experiments, we established that C-to-U and U-to- Ψ conversion happens at the same position.

In the current paper, we conducted an *in vitro* editing assay with SI-labelled oxygen (2 Dalton heavier oxygen) (Fig. 4ef and Fig 3ef). First, we observed that SI-label was introduced to uridine by the TrcP NTD during the deamination reaction (Fig 4f). Then, this SI-label was finally observed in pseudouridine (Fig. 3f), indicating that cytidine is converted into pseudouridine. These results confirm that C-to-U and U-to- Ψ conversion happens at the same position *in vitro*.

2. If the authors did total nucleoside analysis of the purified tRNA^{Tyr}, they should provide the LC-MS data showing that only the modifications shown in the tRNA-Tyr cartoon (Fig. 1) are found in their purified tRNA. This will in fact prove its purity.

We did nucleoside analysis in our previous paper using the same tRNA-Tyr (Nat. Chem. Biol., Kimura et al.). We observed only trace amounts of modifications that are not supposed to be in tRNA-Tyr, indicating that our purification method enables isolation of high purity tRNA-Tyr.

Fig. R5 Detected modifications in LC-MS analysis of purified tRNAs.(from Nat. Chem. Biol. 2020 Kimura et al³.) The area values of a modified nucleoside were normalized using the T signal, which is present in all tRNA species as an internal control. The value relative to the maximum number across the nine tRNA species are shown. tRNA-Tyr did not contain most of the modifications that are not supposed to be in it (modifications except for oQ, Q, ms²io⁶A, m¹A, s⁴U, Ψ, D, and T), suggesting low-level contamination of other tRNA species.

3. In the binding assay. It is not clear, why the data goes from little binding to a lot more binding when comparing 2 uM to 5 uM protein (WT). Also, the amount of tRNA used is a formula for non-specific binding. For the binding to be meaningful and to rule out non-specific binding mediated by the putative RNA-binding domain, the authors should at least provide an apparent dissociation constant with proper quantification.

The reviewer raises a valid point. Measuring dissociation constants is useful to assess binding abilities. However, it is hard to measure the dissociation constant between a tRNA modifying enzyme and a tRNA because their binding is transient. We tried to measure dissociation constants using low concentration tRNA. In these conditions, most of the tRNAs were stacked in the wells, likely due to non-specific binding, so we could not measure the dissociate constant. Then, we turned to qualitative assessments of binding between tRNA and tRNA modifying enzymes because this approach has been accepted in several studies when adequate controls are used⁴⁻⁶. We observed the signals reflecting the complex of tRNA-Tyr and TrcP in Fig. 4c. To exclude the

possibility of non-specific binding (the reviewer's concern), a control experiment was carried out in which TrcP was mixed with another tRNA species, tRNA-Asp. We did not observe the complex signal in this control setting, suggesting that the observed signals reflect specific binding between TrcP and tRNA-Tyr. Additionally, we did not observe the signals of the TrcP-tRNA complex when we used the KR mutant. This result is consistent with our claim that the positively charged patch facilitates specific tRNA binding.

4. When making the claim that tRNA without Q or for that matter *ms²io⁶A* influence pseudouridine formation etc. The authors should provide evidence that the tRNAs purified from those mutants indeed lack those modifications. It has been my experience that several so-called deletion mutants are not quite mutated when tested carefully.

The reviewer raises a valid point. It is crucial to confirm the validity of the mutant strains. We confirmed that the mutants lost the modifications with LC-MS analysis (Fig. R6)

Fig. R6 LC-MS analysis of tRNA-Tyr purified from WT and the indicated modification deletion strains. *ms²io⁶A* and *io⁶A* were eliminated in the $\Delta miaA$ strain, whereas *ms²io⁶A* was converted into *io⁶A* in the $\Delta miaB$ strain. *Q* and its precursor *oQ* were lacking in the Δtgt strain. The biggest peak in each modified nucleoside was adjusted to 100% among strains. *m⁵U* is a control that is supposed to be present in all strains.

5. In the dual reporter assays, an important control is missing. They should provide a similar control for reading U and C-ending codons for tRNAs that do not undergo C to pseudouridine, if the claim is that this effect is specific for the editing and modification in question.

The reviewer raises a good point. It is possible that C-to- Ψ editing and *Q* modification have secondary non-specific effects on general frameshift efficiency. To exclude this possibility, in the revised manuscript, we developed a dual luciferase reporter system, enabling high-throughput analyses of decoding activities. These experiments established the growth phase- and codon-dependent effects of C-to- Ψ editing on Tyr codon decoding. As suggested by the reviewer, we also constructed another set of reporters, testing the Ile-AUC codon, and observed no significant differences in the AUC decoding ability between WT and mutant strains, supporting the idea that C-to- Ψ editing and *Q* modification affect Tyr codons specifically. This data is shown in the revised manuscript in a new Figure 6b.

Fig. R7 Dual luciferase *r* assays evaluating decoding of Tyr and Ile codons. *R_t* and *R_s* represent the F/R values (the signal ratio of Firefly luciferase to Renilla luciferase) of an inframe construct and frameshift construct, respectively. We presume that frameshift and inframe decoding kinetically compete against each other; thus, decoding efficiency is supposed to be inversely correlated with frameshift frequencies. Therefore, a low *R_t/R_s* value (an inversed value of frameshift frequency normalized by inframe translation) is interpreted as low decoding efficiency. *R_t/R_s* values normalized by the WT values are indicated. The top and bottom panels show the results derived from log and stationary phase cultures, respectively. The tested codons are shown above, and the tested strains are labeled below. Ordinary one-way ANOVA was used for a statistical test. Comparisons were made in all combinations among the strains, and Tukey was used for correction for multiple comparisons (***p*<0.01; ****p*<0.001; *****p*<0.0001).

6. The authors should explain why no data is presented on the use of a transcript to test the activities described. If this was mentioned on the paper, I have missed and should therefore discuss/emphasized.

Fig 5a shows that a tRNA transcript generated by *in vitro* transcription is a poor substrate of TrcP. This observation is noted on p9 line 272 of the revised manuscript. The requirement of methylthio group at position 37 for C-to-Ψ editing at least partly explains this observation.

Reviewer #3 (Remarks to the Author):

In this manuscript by Kimura et al., the authors test the enzymatic activity of a putative tRNA modification enzyme in *Vibrio cholerae*. Through a series of biochemical, molecular and genetic experiments, the authors show that tRNA C-to-Psi conversion (TrcP) protein catalyzes both cytosine deamination and pseudouridylation. The authors then use mutagenesis and truncation analysis to demonstrate that the deaminase domain alone is necessary and sufficient for deamination while the pseudoridylase domain requires linkage to the deaminase domain for activity. Moreover, the paper finds that TrcP activity is stimulated by another modification at

position 37 of the tRNA while TrcP-dependent pseudouridylation at position 32 affects the levels of queuosine modification at position 34. Finally, experiments using frameshifting reporter assays suggest that pseudouridine modification impacts decoding of UAU codons in coordination with the queuosine modification. The experiments and assays are well-designed and executed with appropriate controls and standards.

Overall, the results from this paper demonstrate conclusively that TrcP is a tRNA modification enzyme with two distinct activities. These results are noteworthy because they provide insight into a novel molecular mechanism by which cytosine is converted to pseudouridine in tRNA. Since this is the first instance of a single enzyme that catalyzes both editing and pseudouridylation, these findings will be impactful for the RNA field and for future identification of similar enzymes. Based upon the insight provided by these findings, this manuscript is recommended for acceptance with only minor questions and edits noted below.

We thank the reviewer for the positive remarks.

Points

- The use of "editing" when referring to tRNA modifications can be tricky. Due to historical reasons, some in the field might reserve the term "editing" for only C to U and A to I modifications in mRNA. One suggestion to avoid this potential issue is to replace "editing" with "modification". Instead of referring to the TrcP activity as "editing", the Authors could refer to cytosine to uridine conversion as deaminase activity since that is the specific biochemical activity catalyzed by TrcP. Moreover, the title could read "Sequential action of a tRNA modification enzyme in conversion of cytidine to pseudouridine".

The reviewer raises a valid point. Historically, C-to-U and A-to-I conversions were initially described in mRNA; subsequently, the same chemical conversions were discovered in tRNAs. However, to our knowledge, it is well accepted that C-to-U and A-to-I conversion in tRNA have been referred to as "editing" in the literature⁷⁻⁹. In our view, RNA "modification" results in the generation of modified nucleosides that are distinct from the canonical four nucleosides. However, the C-to-U conversion yields a canonical nucleoside, so we think it is more accurate to refer to this process as editing rather than modification. Since TrcP's activity includes the C-to-U editing process, we think it is valid to refer C-to- Ψ conversion as RNA editing.

- Page 3, line 54. The authors state that "However, with some exceptions, the functions and biosynthesis pathways of editing in tRNAs are largely unclear". This statement makes it seem that very little is known about the proteins that catalyze C to U or A to I conversion in tRNAs. However, the biosynthetic pathways for tRNA editing are known for several organisms. We suggest that the authors soften the wording in this sentence.

Thank you for this suggestion. We softened the sentence, "However, the functions and biosynthesis pathways of tRNA editing have not been well characterized outside of a few model organisms."

- In Figure 4, the TrcP mutant in the long helical domain consists of 6 mutations from positively charged residues to alanine. The resulting KRmut protein is unable to rescue modification in cells nor bind tRNA-Tyr. However, the mutation of 6 residues could have rather drastic consequences on protein folding that are not directly linked to tRNA binding. Did the authors test single point mutations in the long helical domain and did they have any influence on activity or tRNA binding? If so, we recommend that the authors include this information, even if the single point mutations had very little effect on activity or RNA binding.

The reviewer raises a valid point that is also addressed in our response to Rev 1, pt 3 above. The six substitutions in the KR mutant could affect protein folding. As discussed above, we found that the melting temperature of the WT and KR mutant were almost the same (Fig. R1), suggesting that six mutations do not drastically affect protein folding. Additionally, we created and tested the editing activity of multiple single alanine substitution mutants;

these assays revealed that even these single mutations diminish TrcP editing activity (Fig. R2), supporting the claim that single residues in the positively charged patch in the LHD support editing.

- It is intriguing that the NTD containing the cytidine deaminase domain can be fused with a different pseudouridylase to yield an active enzyme in vitro. Did the authors test whether this NTD-RluA fusion protein can rescue modification in vivo?

This is an interesting experiment for our future studies -thank you.

- The experimental setup for the translation frameshift assay is unclear. In the Methods, it says "The WT, trcP, tgt, and trcP/ tgt strains were transformed with the frameshift and 0-frame reporters." Does this mean that each strain was transformed with both the frameshift and 0-frame reporters and then fluorescence was measured? Or was each strain transformed with either the frameshift or 0-frame reporters and then each strain containing an individual reporter plasmid was measured? Does each point for each strain represent an independent culture that was measured by FACs or does each point represent technical replicates where the same sample is being measured three times? Please clarify and provide additional details in the manuscript.

Thank you for these comments. As described above, we developed a new dual-luciferase assay system to assess decoding abilities. The details of the new assay are provided in the revised manuscript. See response to Rev. 2 pt 5 above.

- 1 Zhou, W., Richmond-Buccola, D., Wang, Q. & Kranzusch, P. J. Structural basis of human TREX1 DNA degradation and autoimmune disease. *Nat Commun* **13**, 4277, doi:10.1038/s41467-022-32055-z (2022).
- 2 Hemphill, W. O., Salsbury, F. R. & Perrino, F. W. Towards a new model for the TREX1 exonuclease. *bioRxiv*, doi:<https://doi.org/10.1101/2022.02.25.481063> (2022).
- 3 Kimura, S., Dedon, P. C. & Waldor, M. K. Comparative tRNA sequencing and RNA mass spectrometry for surveying tRNA modifications. *Nat Chem Biol* **16**, 964-972, doi:10.1038/s41589-020-0558-1 (2020).
- 4 Sakai, Y., Kimura, S. & Suzuki, T. Dual pathways of tRNA hydroxylation ensure efficient translation by expanding decoding capability. *Nat Commun* **10**, 2858, doi:10.1038/s41467-019-10750-8 (2019).
- 5 Kimura, S. *et al.* Discovery of the beta-barrel-type RNA methyltransferase responsible for N6-methylation of N6-threonylcarbamoyladenine in tRNAs. *Nucleic Acids Res* **42**, 9350-9365, doi:10.1093/nar/gku618 (2014).
- 6 Ikeuchi, Y., Kitahara, K. & Suzuki, T. The RNA acetyltransferase driven by ATP hydrolysis synthesizes N4-acetylcytidine of tRNA anticodon. *EMBO J* **27**, 2194-2203, doi:10.1038/emboj.2008.154 (2008).
- 7 Randau, L. *et al.* A cytidine deaminase edits C to U in transfer RNAs in Archaea. *Science* **324**, 657-659, doi:10.1126/science.1170123 (2009).
- 8 Dixit, S., Henderson, J. C. & Alfonzo, J. D. Multi-Substrate Specificity and the Evolutionary Basis for Interdependence in tRNA Editing and Methylation Enzymes. *Front Genet* **10**, 104, doi:10.3389/fgene.2019.00104 (2019).
- 9 Macbeth, M. R. *et al.* Inositol hexakisphosphate is bound in the ADAR2 core and required for RNA editing. *Science* **309**, 1534-1539, doi:10.1126/science.1113150 (2005).

REVIEWERS' COMMENTS

Reviewer #1 (Remarks to the Author):

The authors have adequately addressed the concerns raised in my original review.

Reviewer #2 (Remarks to the Author):

The authors have fully addressed all my concerns and have made important clarification.

This in my opinion is now a really nice and well-presented story.

Reviewer #3 (Remarks to the Author):

This revised manuscript has addressed the major points from the first review. In response to similar comments raised by two different reviewers, the Authors provide evidence that the 6x mutant TrcP protein is not grossly misfolded or degraded in cells. Moreover, the Authors generated TrcP variants with only single point mutations and observed similar results as the 6x mutant. The Authors have also devised a frameshift reporter system to test the impact of the tRNA modification on translation. These additional experiments strengthen the key findings in the manuscript. As mentioned in the first review, the results are impactful in the RNA modification field by providing insight into a novel molecular mechanism by which cytosine is converted to pseudouridine in tRNA. Based upon the significance of these findings, we recommend this manuscript for acceptance.